# Imprinted *Cdkn1c* genomic locus cell-autonomously promotes cell survival in cerebral cortex development

Susanne Laukoter[1,3], Robert Beattie [1,3], Florian M. Pauler [1,3], Nicole Amberg[1], Keiichi I. Nakayama[2] & Simon Hippenmeyer [1*]

The cyclin-dependent kinase inhibitor p57$^{KIP2}$ is encoded by the imprinted *Cdkn1c* locus, exhibits maternal expression, and is essential for cerebral cortex development. How *Cdkn1c* regulates corticogenesis is however not clear. To this end we employ Mosaic Analysis with Double Markers (MADM) technology to genetically dissect *Cdkn1c* gene function in corticogenesis at single cell resolution. We find that the previously described growth-inhibitory *Cdkn1c* function is a non-cell-autonomous one, acting on the whole organism. In contrast we reveal a growth-promoting cell-autonomous *Cdkn1c* function which at the mechanistic level mediates radial glial progenitor cell and nascent projection neuron survival. Strikingly, the growth-promoting function of *Cdkn1c* is highly dosage sensitive but not subject to genomic imprinting. Collectively, our results suggest that the *Cdkn1c* locus regulates cortical development through distinct cell-autonomous and non-cell-autonomous mechanisms. More generally, our study highlights the importance to probe the relative contributions of cell intrinsic gene function and tissue-wide mechanisms to the overall phenotype.

[1] Institute of Science and Technology Austria, Am Campus 1, 3400 Klosterneuburg, Austria. [2] Department of Molecular and Cellular Biology, Kyushu University, Fukuoka 812-8582, Japan. [3]These authors contributed equally: Susanne Laukoter, Robert Beattie, Florian M. Pauler. *email: simon. hippenmeyer@ist.ac.at

The developmental programs regulating the generation of cortical projection neurons by radial glia progenitor (RGP) cells need to be precisely implemented and regulated[1–3]. Impairments in neuron production and maturation lead to alterations in the cortical cytoarchitecture which is thought to represent the major underlying cause for a range of neurodevelopmental disorders including microcephaly, megalencephaly, epilepsy and autism spectrum disorders[4]. However, the cellular and molecular mechanisms that underlie concerted RGP lineage progression and the control of neuron output remain unclear[3,5,6]. Here we genetically dissect the ill-defined function of the essential *Cdkn1c* gene in corticogenesis. Previous studies indicate that *Cdkn1c*, encoding p57[KIP2], acts as growth/tumor suppressor by directly inhibiting the activity of cyclin-dependent kinases (CDKs)[7–10]. The *Cdkn1c* genomic locus is subject to genomic imprinting resulting in the expression of the maternal and silencing of the paternal allele, respectively[11,12]. Genetic loss of function studies indicate an important role of p57[KIP2] in regulating RGP lineage progression and cortical projection neuron genesis[13,14]. Mutant *Cdkn1c*[−/−] mice exhibit macrocephaly and cortical hyperplasia indicating a critical function in tuning RGP-mediated neuron output, supporting the concept of a growth-inhibitory *Cdkn1c* gene function[14]. However, whether and how *Cdkn1c* regulates RGP proliferation behavior cell-autonomously is not known. Interestingly, brain-specific conditional deletion of *Cdkn1c* using Nestin-Cre driver results in thinning of the cerebral cortex, a phenotype seemingly opposite to the one in global *Cdkn1c* knockout[15]. Thinning of the cortex however likely emerges as an indirect secondary effect due to severe hydrocephalus caused by a defect in the subcommissural organ (SCO) which is required for cerebrospinal fluid flow[15,16]. Thus the function of *Cdkn1c* in corticogenesis may involve substantial non-cell-autonomous components which could promote or inhibit RGP-mediated neuron output and/or neuronal maturation. Here we address this issue and analyze the cell-autonomous phenotypes upon genetic *Cdkn1c* gene ablation at single-cell level by capitalizing on mosaic analysis with double markers (MADM) technology. Our data from MADM-based analysis indicate that the well-established growth-inhibitory *Cdkn1c* function is a non-cell-autonomous effect of *Cdkn1c* knockout in the whole organism. In contrast, we reveal a growth-promoting cell-autonomous *Cdkn1c* function, which at the mechanistic level acts to protect cells from p53-mediated apoptosis. This cell-autonomous *Cdkn1c* survival function is dosage sensitive but not subject to genomic imprinting and is attributed to the genomic *Cdkn1c* genomic locus rather than the expressed *Cdkn1c* transcript.

## Results

**MADM-based analysis of *Cdkn1c* imprinting phenotypes**. In order to determine the degree of cell-autonomy of imprinted *Cdkn1c* gene function in cortical development, we used genetic MADM paradigms[17–19]. To this end, we capitalize on two unique properties of the MADM system: (1) the cell-type-specific generation and visualization of uniparental chromosome disomy (UPD, somatic cells with two copies of the maternal or paternal chromosome) for the functional analysis of imprinted dosage-sensitive gene function; and (2) the sparseness of UPD generation for analyzing cell-autonomous phenotypes at single-cell resolution. Since the imprinted *Cdkn1c* locus, located on mouse chromosome 7 (Chr. 7), exhibits maternal expression[11,12], MADM-labeled cells carrying maternal UPD (matUPD, two maternal chromosomes) are predicted to express two copies of *Cdkn1c* and cells with paternal UPD (patUPD, two paternal chromosomes) would not express *Cdkn1c* (Fig. 1a). Thus, the phenotypic consequences of *Cdkn1c* loss (patUPD) and gain (matUPD) of

function can be assessed simultaneously in MADM-induced UPDs, which also express distinct fluorescent reporters (Fig. 1a). MADM-based generation of Chr. 7 UPD occurs only in a very small fraction of genetically defined cells[18] and permits the analysis of postnatal stages since the sparseness of genetic mosaicism enables the bypassing of early lethality associated with loss of *Cdkn1c* function[10,20].

**No cell-autonomous role for *Cdkn1c* in cortical neurogenesis**. To generate mice with MADM-induced UPD of Chr. 7 (MADM-7) in proliferating cortical RGPs, we crossed female *MADM-7*[TG/TG] with male *MADM-7*[GT/GT];*Emx1*[Cre/+] (Supplementary Figs. 1a and 2a) to obtain experimental *MADM-7*[GT/TG];*Emx1*[Cre/+] mice. In MADM-7 mice, sparse green GFP[+] cells carry matUPD (prediction: two copies of expressed *Cdkn1c*), red tdT[+] patUPD (prediction: no *Cdkn1c* expression), yellow GFP[+]/tdT[+] and unlabeled cells carry no UPD (prediction: 1× *Cdkn1c* expression) (Fig. 1b). In order to assess and validate whether the above MADM paradigm faithfully results in the predicted levels of *Cdkn1c* expression, we isolated cells carrying matUPD and patUPD, and control cells at embryonic day (E) 13 and E16 based on their fluorescence by FACS. Next the samples were processed for RNA sequencing (RNA-seq) and from the obtained sequences we quantified the relative levels of *Cdkn1c* expression in matUPD and patUPD, when compared to control (Fig. 1c, Supplementary Data 1, 2). As predicted from the MADM scheme the levels of *Cdkn1c* expression were (1) much lower in patUPD in comparison to matUPD; and (2) while *Cdkn1c* expression was nearly absent in patUPD at E16, the level in matUPD (two copies of *Cdkn1c*) reached approximately twofold of the level in control (one copy of *Cdkn1c*) (Fig. 1c). To further assess the experimental UPD paradigm and to corroborate the above results, we generated comprehensive coverage plots for the RNA reads in the *Cdkn1c* genomic locus (Supplementary Fig. 3a–c, f) and the larger *Kcnq1* cluster region (Supplementary Fig. 4a–c, f). The in-depth analysis of these coverage plots revealed (1) no novel, previously unannotated, transcripts in the *Cdkn1c* locus; and (2) that the predicted *Cdkn1c* expression occurs faithfully in UPD as predicted from the MADM scheme (Fig. 1a). Specifically, cells with matUPD (two doses of *Cdkn1c*; red in Fig. 1c and Supplementary Figs. 3a and 4a) show about twofold higher *Cdkn1c* expression levels than control cells (one dose of *Cdkn1c*; yellow in Fig. 1a and dark yellow in Supplementary Figs. 3c and 4c) whereas cells with patUPD (zero dose of *Cdkn1c*; green in Fig. 1a and blue in Supplementary Figs. 3b and 4b) show a drastic reduction of *Cdkn1c* expression. Thus, these results conclusively validate the MADM approach resulting in distinct doses of expressed *Cdkn1c* transcripts depending on the UPD status.

Since *Cdkn1c* promotes growth inhibition and cell-cycle exit, and taking into consideration the imprinting and expression status of *Cdkn1c*, green matUPD (two doses of *Cdkn1c*) cells would be expected to show growth/proliferation disadvantage when compared to red patUPD (no *Cdkn1c*) cells (Fig. 1a). Perhaps surprisingly we however found a green/red (G/R), i.e. matUPD/patUPD ratio of ~1 and thus equal growth/proliferation potential of RGPs regardless of the UPD status (Fig. 1d–f). These results indicate that *Cdkn1c* has no prominent cell-autonomous role in cell-cycle regulation of proliferating cortical RGPs. To directly test this possibility, we first analyzed the expression levels of cell-cycle regulators in RNA-seq data. We could not find significant differences in the expression levels of a number of key cell-cycle regulators in cells with matUPD or patUPD, respectively (Fig. 2a, b, Supplementary Data 2). Next, we measured incorporation of EdU after 1 h of exposure at E13 in the embryonic neuroepithelium and found that MADM-labeled cells with matUPD, patUPD and control all displayed equal relative

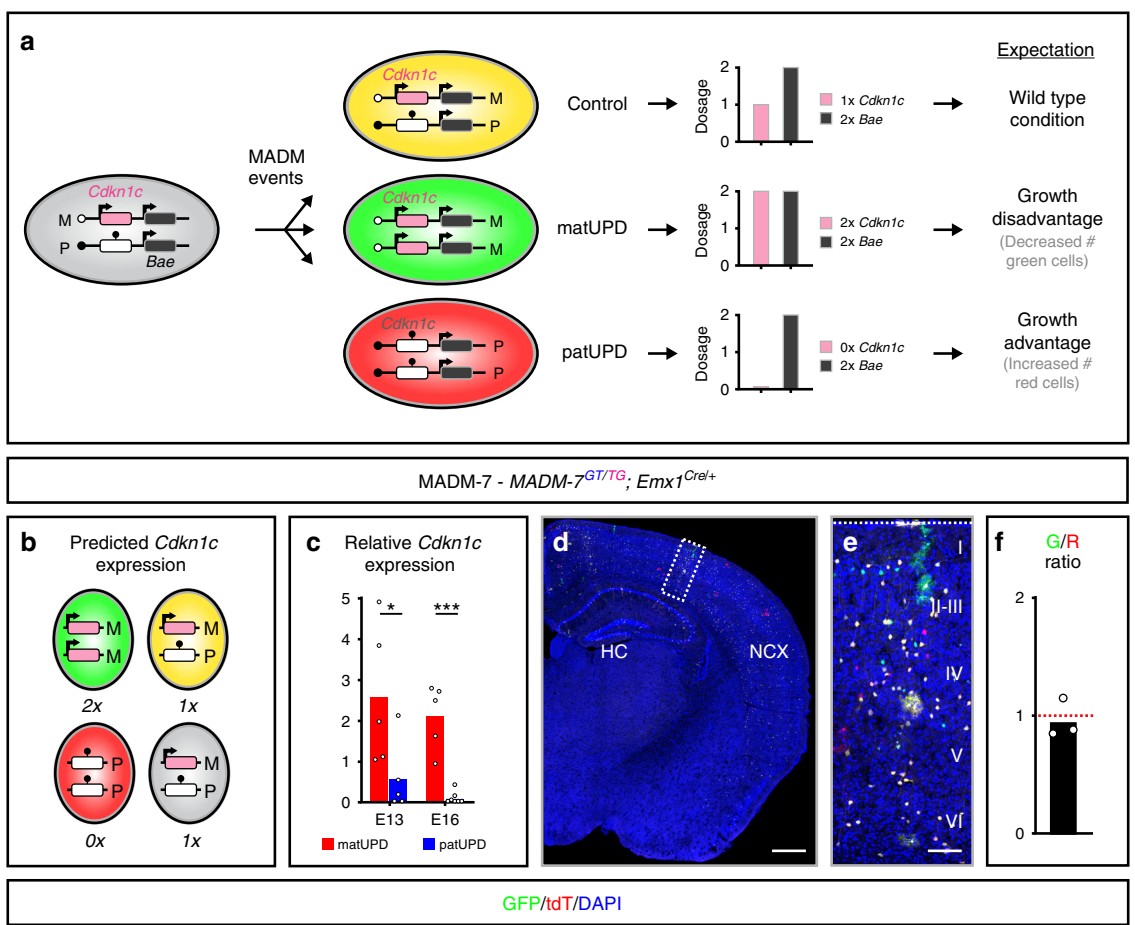

**Fig. 1 MADM-based analysis of imprinted *Cdkn1c* gene function at single-cell level. a** MADM recombination events result in distinct fluorescent labeling of cells containing uniparental disomy (UPD). Yellow cells are control cells, green cells carry maternal uniparental chromosome disomy (matUPD) and red cells contain paternal uniparental chromosome disomy (patUPD). *Cdkn1c* is expressed from the maternal allele in yellow cells, which resembles the wild-type situation. In green cells (matUPD) *Cdkn1c* is expressed from both maternal alleles and predicted to result in growth/proliferation disadvantage (expression of two doses of a growth inhibitor). Red cells (patUPD) lack *Cdkn1c* expression and are expected to show growth/proliferation advantage due to lack of expression of a growth suppressor. **b** Schematic depicts green (GFP+), red (tdT+), yellow (GFP+/tdT+) and unlabeled MADM cells with UPD (red, green) and control cells (yellow, unlabeled). Parental origin of chromosome is indicated (M, maternal; P, paternal). Imprinting status of *Cdkn1c* (arrow, expression; ball on stick, repression) and predicted expression (0×, 1×, 2×) is indicated. **c** Relative *Cdkn1c* expression in matUPD (red bars) and patUPD (blue bars) at E13 and E16. Bars represent mean. *$p < 0.05$, ***$p < 0.001$ (Wald test). Data points indicate individual animals ($n = 4$–7). **d** MADM-labeling pattern in cerebral cortex of MADM-7 (*MADM-7*^GT/TG^;*Emx1*^Cre/+^) at P21. The parent from which the MADM cassettes were inherited is indicated in the respective genotypes in pink (mother) and blue (father). **e** Higher magnification of cortical cross-section (boxed area in (**d**)). **f** G/R ratio of single MADM-labeled cortical neurons is depicted as geometric mean. Note the equipotency of cells with matUPD and patUPD. Nuclei (**d**, **e**) were stained using 4′,6-diamidino-2-phenylindole (DAPI, blue). Data points indicating individual animals ($n = 3$). Source data are provided as a Source Data file. Scale bar, 500 μm (**d**), 90 μm (**e**).

amounts with EdU label (Fig. 2c, d). Lastly, we monitored the fraction of proliferating cells that remained in cell-cycle (EdU+/Ki67+) in a 24-h time window. Again, we found similarly sized EdU+/Ki67+ fractions of MADM-labeled cells with matUPD, patUPD and control (Fig. 2e, f). Altogether these data indicate that individual RGPs with matUPD (two expressed doses of *Cdkn1c*) or patUPD (no expressed *Cdkn1c*), in an otherwise wild-type environment, exhibit similar proliferation behaviors. Based on the sparse induction of UPD in just very few cells (with a vastly normal background), these results show that *Cdkn1c* does not regulate RGP proliferation behavior cell-autonomously. This is also in agreement with the observed sparse expression pattern of p57^KIP2 protein in RGPs (less than 12% of PAX6+ RGPs express p57^KIP2)[14]. The results thus indicate that the observed cortical overgrowth observed in *Cdkn1c*^−/− full knockout mice[14] is mainly due to global non-cell-autonomous *Cdkn1c* function and/or community effects.

**No loss of imprinted *Cdkn1c* expression in cortical cells**. An alternative explanation for our results indicating equipotency of Chr. 7 matUPD/patUPD could be the possible cell-type-specific loss of imprinting, a phenomenon which has been observed for imprinted *Dlk1* in postnatal stem cells[21]. In other words, the paternal *Cdkn1c* allele would be de-repressed specifically in *Emx1*+ RGPs and/or their lineage. Thus cells with mat- and patUPD would both express similar doses of *Cdkn1c* and therefore show the same phenotype, i.e. the same number of projection neuron output from RGPs. To test this possibility we generated F1 C57BL/6J (B6)—CAST/EiJ (CAST) hybrids[22] to qualitatively and quantitatively analyze allelic expression in RGPs and nascent neurons. We used two well-defined single nucleotide polymorphisms (SNPs) located in exon 2 and exon 4 of *Cdkn1c* (Supplementary Fig. 5a), and a SNP in *Ndn* (single exon gene which is paternally expressed) as control[22]. We first isolated genomic DNA from an individual B6/CAST hybrid embryo at

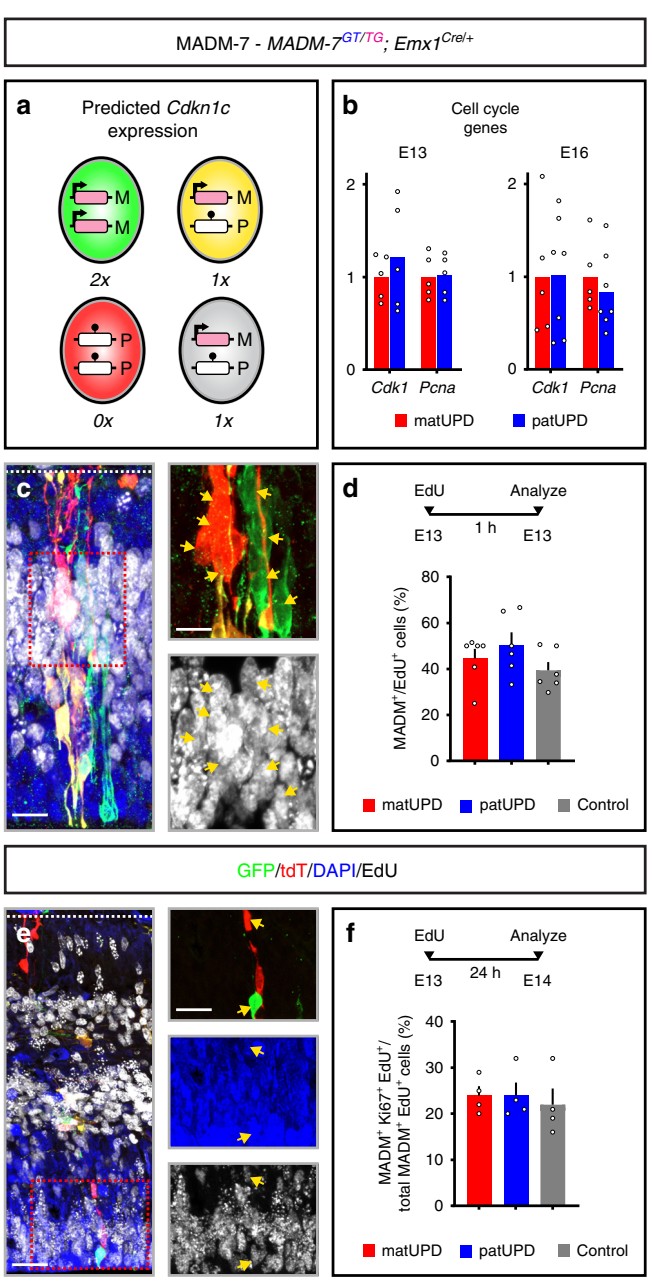

**Fig. 2 Analysis of imprinted *Cdkn1c* cell-cycle properties based on MADM-UPD. a** Schematic depicts green (GFP⁺), red (tdT⁺), yellow (GFP⁺/tdT⁺) and unlabeled MADM cells with UPD (red, green) and control cells (yellow, unlabeled). Parental origin of chromosome is indicated (M, maternal; P, paternal). Imprinting status of *Cdkn1c* (arrow, expression; ball on stick, repression) and predicted expression (0×, 1×, 2×) is indicated. **b** Expression levels of two representative cell-cycle genes (*Cdk1*, *Pcna*) in matUPD (red bars) and patUPD (blue bars) at E13 and E16. Data points indicate individual animals ($n = 5$−7). **c** EdU labeling (1 h chase) in MADM-7 (*MADM-7^GT/TG;Emx1^Cre/+*) cerebral cortex at E13 with GFP⁺ matUPD (green) and tdT⁺ patUPD (red) (top) and colabeled with EdU in white (bottom) double-positive cells are indicated by yellow arrows. **d** Fraction (%) of MADM⁺/EdU⁺ colabeled cells (red bar, matUPD (GFP⁺ cells); blue bar, patUPD (tdT⁺ cells); gray bar, control cells (GFP⁺/tdT⁺ cells)). Data points indicate individual animals ($n = 6$). **e** EdU⁺/Ki67⁺ colabeling in MADM-7 (*MADM-7^GT/TG;Emx1^Cre/+*) cerebral cortex with EdU injection (24 h chase) at E13. MADM-labeled (matUPD, GFP⁺; patUPD, tdT⁺ in top) Ki67⁺ (blue in middle) and EdU⁺ (white in bottom) triple-positive cells are indicated by yellow arrows. **f** Fraction (%) of MADM⁺/Ki67⁺/EdU⁺ cells of total MADM⁺/EdU⁺ cells after 24 h EdU chase (red bar, matUPD (GFP⁺); blue bar, patUPD (tdT⁺); gray bar, control cells (GFP⁺tdT⁺)). Data points indicate individual animals ($n = 4$). All bars represent mean. Error bars represent SEM (**d**, **f**). Nuclei (**c**) were stained using DAPI (blue). NCX neocortex, HC hippocampus. Scale bar, 15 μm (**c**), 10 μm (**c** top and bottom), 25 μm (**e**), 20 μm (**e** top, middle, bottom). Source data are provided as a Source Data file.

from allelic expression experiments corroborate the above findings of MADM-based analysis of Chr. 7 UPD (Fig. 1). Together, these data demonstrate that *Cdkn1c* does not cell-autonomously control RGP-mediated neuron output and/or maturation, and that the observed macrocephaly in *Cdkn1c⁻/⁻* full knockout likely reflects global organism overgrowth.

**Genetic *Cdkn1c* ablation results in microcephaly**. To more directly assess *Cdkn1c* function in cortical RGPs at single-cell resolution, we next introduced a conditional *Cdkn1c-flox* allele[24] into MADM-7 (*MADM-7^GT/TG,Cdkn1c-flox;Emx1^Cre/+*). When we introduced the *Cdkn1c-flox* allele from the mother, all cells (regardless of UPD status) of the *Emx1⁺* lineage in F1 should in principle be equivalent to homozygous *Cdkn1c⁻/⁻* because of imprinting, i.e. deleted maternal expression and paternal silencing (Fig. 3a and Supplementary Figs. 1b and 2b). Indeed, we found that control cells, and cells with matUPD and patUPD, all displayed nearly undetectable relative *Cdkn1c* expression levels (Fig. 3b). However, contrary to our expectation of a cortical overgrowth phenotype due to *Cdkn1c* loss of function, and RGP equipotency with G/R of 1, we observed a dramatic reduction of GFP⁺ *Cdkn1c⁻/⁻* matUPD cells when compared to tdT⁺ *Cdkn1c⁺/⁺* patUPD (although with two copies of silenced *Cdkn1c*) and severe microcephaly (Fig. 3c–f). This *Cdkn1c* conditional deletion phenotype is in stark contradiction with a growth suppressive function of *Cdkn1c*. Next, we evaluated whether the suspected growth-promoting function of *Cdkn1c* is subject to regulation by genomic imprinting. To this end we introduced the *Cdkn1c-flox* allele from the father (Fig. 3g and Supplementary Figs. 1b and 2c) and generated GFP⁺ *Cdkn1c⁻/⁻* patUPD cells which we compared to tdT⁺ *Cdkn1c⁺/⁺* matUPD. Analysis of the relative *Cdkn1c* expression confirmed that patUPD express very low levels of *Cdkn1c* while matUPD (with two copies of *Cdkn1c*) show about twofold higher *Cdkn1c* expression in comparison to control cells (one copy of *Cdkn1c*) (Fig. 3h). Strikingly we observed again dramatic reduction of mutant *Cdkn1c⁻/⁻* cells when compared to *Cdkn1c⁺/⁺* cells (Fig. 3i–l). This is unexpected because the

E12 to confirm the presence of respective SNPs in *Cdkn1c* and *Ndn*. Sanger and deep sequencing of the genomic DNA confirmed the presence and ~50% abundance of each parental SNP as expected (Supplementary Fig. 5b–h, o). Next, we generated single-cell suspensions of E12 cortex from B6/CAST hybrids and used well-established FACS protocols[23] to enrich for RGPs and projection neurons. From the obtained neuron and progenitor populations, we isolated RNA which we converted to cDNA, followed by Sanger and deep sequencing to determine allelic expression (Supplementary Fig. 5b–q). These experiments revealed highly skewed, almost exclusive allelic expression of paternally expressed *Ndn*, and maternally expressed *Cdkn1c* in both RGP and neuron cell populations. Importantly, paternal expression of *Cdkn1c* in neurons (<5%) and progenitors (<2%) was miniscule (Supplementary Fig. 5p, q). Thus imprinting and repression of the paternal *Cdkn1c* allele is intact in embryonic neurogenic RGPs and nascent neurons. In conclusion, the results

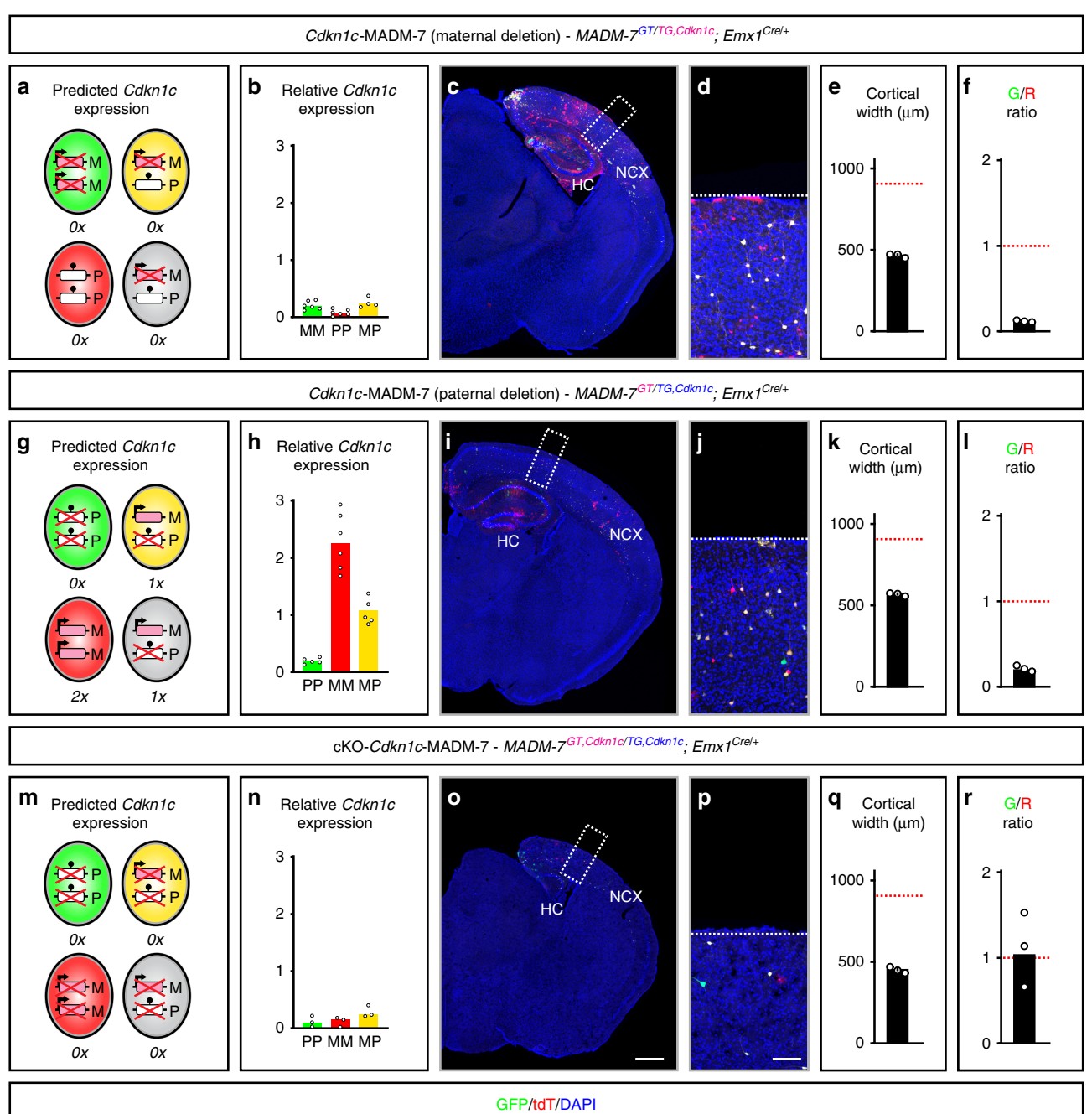

**Fig. 3 Cell-autonomous growth-promoting *Cdkn1c* function is independent of genomic imprinting status.** Analysis of *Cdkn1c*-MADM-7 (*MADM-7^GT/TG, Cdkn1c;Emx1^Cre/+*) with maternal deletion (**a–f**), *Cdkn1c*-MADM-7 (*MADM-7^GT/TG,Cdkn1c;Emx1^Cre/+*) with paternal deletion (**g–l**), and cKO-*Cdkn1c*-MADM-7 (*MADM-7^GT,Cdkn1c/TG,Cdkn1c;Emx1^Cre/+*) (**m–r**). The parent from which the MADM cassettes with or without recombined *Cdkn1c-flox* allele (*Cdkn1c*) were inherited is indicated in the respective genotypes in pink (mother) and blue (father). Predicted *Cdkn1c* expression in GFP+ (green), tdT+ (red), GFP+/tdT+ (yellow) and unlabeled (gray) MADM-labeled cells (**a**, **g**, **m**); and true relative levels of *Cdkn1c* expression in GFP+ (green bar), tdT+, (red bar), GFP+/tdT+ (yellow bar) MADM-labeled cells (**b**, **h**, **n**) in *Cdkn1c*-MADM-7 with maternal deletion (**a**, **b**), *Cdkn1c*-MADM-7 with paternal deletion (**g**, **h**), and cKO-*Cdkn1c*-MADM-7 (**m**, **n**) at E16 are indicated (MM, matUPD; PP, patUPD; MP, control). Data points indicate individual animals (*n* = 5–7). Outlier values are not shown (see Source Data File). Bars indicate median. Imprinting of *Cdkn1c* (arrow, expression; ball on stick, repression) and predicted expression (0×, 1×, 2×) is indicated. Parental origin of chromosome is indicated (M, maternal; P, paternal) and conditional deletion of *Cdkn1c* is marked with red cross. MADM-labeling (GFP+, green; tdT+, red; GFP+/tdT+, yellow) (**c**, **d**, **i**, **j**, **o**, **p**), cortical width (μm) is shown with an error bar representing SEM (**e**, **k**, **q**), and G/R ratio (geometric mean) of single MADM-labeled cortical projection neurons (**f**, **l**, **r**) in *Cdkn1c*-MADM-7 with maternal deletion (**c–f**), *Cdkn1c*-MADM-7 with paternal deletion (**i–l**), and cKO-*Cdkn1c*-MADM-7 (**o–r**) at P21 are indicated. Boxed areas in overview images (**c**, **i**, **o**) show representative images of the extent of microcephaly at higher resolution (**d**, **j**, **p**). Nuclei were stained using DAPI (blue). Scale bar, 500 μm (overview (**c**, **i**, **o**)) and 90 μm (inset (**d**, **j**, **p**)). Note the decreased numbers of *Cdkn1c^−/−* cells when compared to *Cdkn1c^+/+* cells in (**f**) and (**l**) but equipotency/equal numbers of *Cdkn1c^−/−* cells in (**r**) regardless of the UPD status. Data points (**e**, **f**, **k**, **l**, **q**, **r**) indicating individual animals (*n* = 3). NCX neocortex, HC hippocampus. Source data are provided as a Source Data file.

imprinting status of *Cdkn1c* in patUPD already shows no expression of *Cdkn1c* due to homozygosity and complete silencing of both paternal *Cdkn1c* alleles (Fig. 3g, h). To further support our finding of a *Cdkn1c* growth-promoting function independent of parental Chr. 7 UPD status, we introduced the *Cdkn1c-flox* allele from both parents and generated a true *Cdkn1c* conditional knockout (cKO) but with sparse MADM-labeling for single-cell analysis (MADM-7$^{GT,Cdkn1c\text{-}flox/TG,Cdkn1c\text{-}flox}$;*Emx1*$^{Cre/+}$) (Fig. 3m and Supplementary Figs. 1c and 2d). In these cKO-*Cdkn1c*-MADM-7 mice control cells and cells with matUPD and patUPD all displayed nearly undetectable levels of *Cdkn1c* expression (Fig. 3n) similar to the paradigm with maternal *Cdkn1c* deletion (Fig. 3b). Consequently, the cKO-*Cdkn1c*-MADM-7 mice exhibit very strong microcephaly (Fig. 3o–q), similar to *Cdkn1c*-MADM-7 mice with maternal deletion (Fig. 3c–e), and slightly more severe than with paternal deletion (Fig. 3i–k). Importantly however, in cKO-*Cdkn1c*-MADM-7 mice we observed a G/R of ~1 unlike the very low G/R ratio in *Cdkn1c*-MADM-7 with maternal or paternal deletion, respectively (Fig. 3f, l, r). A G/R ratio of ~1 in cKO-*Cdkn1c*-MADM-7 cortex indicates equipotency of mutant *Cdkn1c*$^{−/−}$ neurogenic RGP cells regardless of the disomy status (i.e. green GFP$^+$ are patUPD and red tdT$^+$ are matUPD) similar like in MADM-7 where all cells are *Cdkn1c*$^{+/+}$ although with UPD (Fig. 1).

Next, we analyzed the emergence of the microcephaly phenotype during development in a time course analysis. We found that the microcephaly phenotype was already evident from E13 onwards in *Cdkn1c*-MADM-7 with maternal and paternal deletion, respectively, and in cKO-*Cdkn1c*-MADM-7. The severity of microcephaly increased until E16 (Supplementary Fig. 6), persisted until birth and in postnatal mice up to P21 (latest time point of analysis).

In summary, our MADM-based analysis of UPD and in combination with conditional deletion of *Cdkn1c* revealed a growth-promoting function of *Cdkn1c* which is dominant over the imprinting status of the genomic *Cdkn1c* locus. Given that mosaic *Cdkn1c*-MADM-7 mice (all cells are *Cdkn1c*$^{flox/+}$) with heterozygous *Cdkn1c* deletion in *Emx1*$^+$ lineage show microcephaly similar like cKO-*Cdkn1c*-MADM-7 mice, we conclude that the growth-promoting function of *Cdkn1c* is also highly dosage sensitive. In other words, removal of one copy of *Cdkn1c* leads to haploinsufficiency with nearly identical phenotype to that observed when both copies were ablated.

**Gene expression profile upon genetic deletion of *Cdkn1c*.** To gain better insight into the putative mechanism of the growth-promoting *Cdkn1c* function, we first profiled global gene expression in cKO and compared to control mice. To this end we purified cells of the *Emx1*$^+$ lineage in E16 cortices from control MADM-7 and cKO-*Cdkn1c*-MADM-7 mice by FACS. Next, the samples were processed for RNA-seq and global gene expression profiles were established for further analysis (Fig. 4a, Supplementary Data 3) (see Methods for details). First, we reduced the dimensionality of our data and identified similarities and differences in global gene expression between control and cKO-*Cdkn1c*-MADM-7 by principal component analysis (PCA). We found that the samples with distinct genotypes (i.e. control vs. cKO) segregated from each other (Fig. 4b). Next, we confirmed the loss of *Cdkn1c* expression in cKO-*Cdkn1c*-MADM-7 (Fig. 4c). To exclude the possibility of any remaining truncated partial *Cdkn1c* mRNA species and/or the residual presence of previously unannotated transcripts within the *Cdkn1c* genomic locus and broader *Kcnq1* cluster region in cKO-*Cdkn1c*-MADM, we generated coverage plots for the obtained RNA reads (Supplementary Figs. 3d–f and 4d–f). This analysis confirmed (1) the drastic

reduction of *Cdkn1c* transcripts in cKO-*Cdkn1c*-MADM and (2) did not reveal any novel unannotated transcripts that could hypothetically escape deletion in our conditional genetic experimental paradigm. These results thus conclusively validate our genetic deletion approach to conditionally ablate *Cdkn1c* expression in cortical *Emx1*$^+$ lineage.

To obtain a first pass measure of the extent of differential gene expression in cKO-*Cdkn1c*-MADM-7, we plotted the number of up- and downregulated genes (Fig. 4d, Supplementary Data 4). Since imprinted *Cdkn1c* is embedded in a larger cluster, the *Kcnq1*-cluster of imprinted genes on Chr.7, we analyzed whether the expression of neighboring genes could be affected in the cKO-*Cdkn1c*-MADM-7 mice with conditional *Cdkn1c* deletion (Fig. 4e). While the expression profile of *Cdkn1c* indicated drastically low, nearly absent, levels of expression, the genes flanking (2 Mbp up- and downstream) the *Cdkn1c* genomic locus displayed no significant differential expression (Fig. 4e). We next performed gene-ontology (GO) enrichment analysis of the differentially expressed genes in cKO-*Cdkn1c*-MADM-7. This analysis revealed a very high probability of cellular phenotypes associated with the downregulation of genes related to neurogenesis in general and with the upregulation of genes involved in cell death in particular (Fig. 4f, Supplementary Data 5).

**Cdkn1c is cell-autonomously required for cellular survival.** Based upon our findings from gene expression profiling and since previous studies showed that the hydrocephalus phenotype in *Cdkn1c* cKO induced with Nestin-Cre can be rescued by concomitant p53 (*Trp53*) ablation[15], we next analyzed cell death parameters upon conditional loss of *Cdkn1c*. We thus stained cortex at E13 in MADM-7, *Cdkn1c*-MADM-7 and cKO-*Cdkn1c*-MADM-7 embryos for apoptotic cells with antibodies against Caspase-3 (Fig. 5). While in MADM-7 almost no cortical cells showed signs of apoptosis (Fig. 5a–e), high numbers (up to 20% of all cells) of Caspase-3$^+$ cells were detected in *Cdkn1c*-MADM-7 (maternal and paternal deletion) and cKO-*Cdkn1c*-MADM-7 (Fig. 5f–t).

Interestingly, at E13 the relative number of homozygous mutant cells, i.e. green *Cdkn1c*$^{−/−}$ cells, in *Cdkn1c*-MADM-7 (maternal and paternal deletion) was already reduced in comparison to red *Cdkn1c*$^{+/+}$ wild-type cells (Fig. 5h, m). Note that again this reduction in cell number appeared irrespective of the chromosomal disomy status and thus independent of genomic imprinting. However, *Cdkn1c*-MADM-7 mice are genetic mosaics with green cells that show homozygous deletion of the *Cdkn1c* genomic locus, red cells with homozygous intact *Cdkn1c* locus and heterozygous yellow/unlabeled cells with one deleted copy of *Cdkn1c* locus. Therefore, we next analyzed whether the red cells (tdT$^+$) with homozygous intact *Cdkn1c* locus in *Cdkn1c*-MADM-7 (maternal and paternal deletion) mice would have a survival advantage or disadvantage when compared to the heterozygous yellow or unlabeled (DAPI$^+$) cells. Strikingly, far fewer red cells with homozygous intact *Cdkn1c* genomic locus were positive for Caspase-3 than unlabeled (DAPI$^+$) heterozygous cells, regardless of the imprinting status and thus expressed dose of *Cdkn1c* transcript (Supplementary Fig. 7). Collectively, these results demonstrate that the genetic deletion of *Cdkn1c* genomic locus results in an increased probability of cell death in a highly dosage-dependent manner. In other words, removal of one copy of *Cdkn1c* results in haploinsufficiency with increased apoptosis when compared to wild-type cells with two intact copies of *Cdkn1c*.

In the course of our analysis of cell death upon genetic deletion of *Cdkn1c*, we noticed that Caspase-3$^+$ cells in *Cdkn1c*-MADM-7

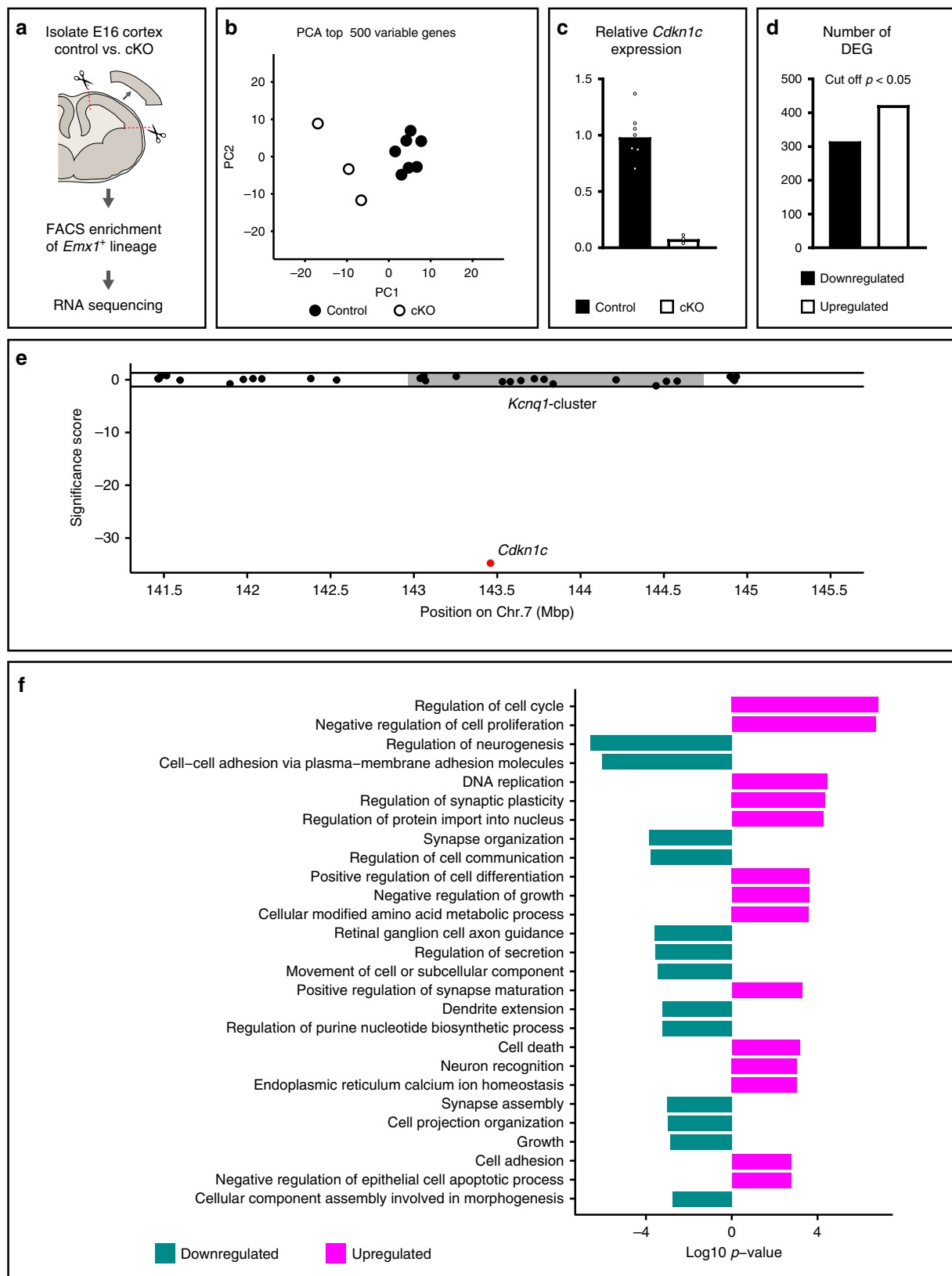

and cKO-*Cdkn1c*-MADM-7 at E13 seemed to appear less prominently in the ventricular zone (VZ) but more abundant in nascent neurons located in the emerging cortical plate (CP). To quantitatively assess the rate of cell death in ventricular RGPs and nascent neurons in the CP, we stained sections from MADM-7,

*Cdkn1c*-MADM-7 and cKO-*Cdkn1c*-MADM-7 embryos with antibodies against Caspase-3 together with the RGP marker PAX6 or nascent neuron marker NEUROD2, respectively. Indeed, the absolute number of PAX6[+]/Caspase-3[+] was significantly lower than the number of NEUROD2[+]/Caspase-3[+]

**Fig. 4 Gene expression profile in cortex upon genetic deletion of *Cdkn1c* locus. a** Experimental paradigm. **b** Principal component analysis of all samples used for analysis. Data points indicate individual animals ($n = 3$–7). **c** Relative levels of *Cdkn1c* in cKO and controls cells. Bars represent mean. Data points indicate individual animals ($n$ (controls) = 7, $n$ (cKO) = 3). **d** Number of up-, and downregulated genes in cKO/control comparisons (adjusted $p$ value < 0.05, Wald test). **e** Significance scores of all informative genes within 2 Mbp up-, and downstream of *Cdkn1c* are shown ($y$ axis). Each dot represents one gene with the position given as the midpoint of its genomic locus (mega base-pairs (Mbp), $x$ axis). Horizontal lines mark limits for significant differential expression (corresponding to an adjusted $p$ value of 0.05, Wald test). Gray box marks the putative maximum size of the *Kcnq1* imprinted gene cluster. **f** Curated list of significantly enriched gene ontology (GO) terms in the lists of up-, downregulated genes shown in (**d**). The full list of significantly enriched GO terms is available in Supplementary Data 5. Source data are provided as a Source Data file.

cells in *Cdkn1c*-MADM-7 and cKO-*Cdkn1c*-MADM-7 (Supplementary Fig. 8). Thus, not only nascent cortical projection neurons (NEUROD2[+]) show a higher incidence of cell death upon *Cdkn1c* ablation but also PAX6[+] RGPs. Consequently, the overall number of actively proliferating RGPs is already reduced from early E13 embryonic stages onward (Supplementary Fig. 9). We also corroborated our findings by staining sections from MADM-7, *Cdkn1c*-MADM-7 and cKO-*Cdkn1c*-MADM-7 embryos with antibodies against p53 which marks cells that initiate the apoptotic pathway (Supplementary Fig. 10). Lastly, both upper layer callosal (SATB2[+]) and deep layer corticofugal (TBR1[+]) projection neuron populations were cell-autonomously prone to apoptosis and thus reduced in *Cdkn1c*-MADM-7 with maternal and paternal deletion, respectively (Fig. 6). Altogether, we conclude that the cell-autonomous growth-promoting *Cdkn1c* function discovered here is mainly acting to promote the survival of differentiating and maturing cortical projection neurons and (to a lesser extent) proliferating RGPs.

## Discussion

Here we report that the *Cdkn1c* genomic locus, rather than the *Cdkn1c* transcript, is cell-autonomously required for the survival of RGPs and nascent projection neurons in the developing cerebral cortex. The reported cell-autonomous *Cdkn1c* survival function is dosage-sensitive but independent of genomic imprinting. Below we discuss the implications of our findings for cortical development but also more generally in the context of *Cdkn1c* imprinting and *Cdkn1c*/p57[KIP2]-mediated growth control at the individual cell- and systemic level.

The development of the cerebral cortex is highly orchestrated and regulated by diverse genetic and epigenetic mechanisms. RGPs are the main progenitor stem cells that produce all excitatory projection neurons[1,3,25] and recent data indicate that cortical neurogenesis follows a relatively deterministic rule[17,26]. The genetic, molecular and cellular mechanisms that regulate RGP lineage progression and nascent projection neuron maturation are however not well understood. Furthermore, recent studies indicate that epigenetic regulatory cues at cellular and systemic levels play an important role as well[5,27]. One peculiar epigenetic regulatory mechanism is genomic imprinting, resulting in the selective silencing of one parental allele in a subset of genes[28–30]. Although many imprinted genes are prominently expressed during corticogenesis, their functional role in regulating cortical development is mostly unclear due to the lack of assays permitting single-cell phenotypic analysis. Here we employed MADM technology to genetically dissect the level of cell-autonomy of imprinted *Cdkn1c* gene function in corticogenesis (Fig. 7). We found that *Cdkn1c* regulates RGP proliferation and cortical neurogenesis predominantly non-cell-autonomously, a concept in agreement with previous studies[14] that implicate *Cdkn1c* in whole organism growth suppression[10] (Fig. 7a, b). In contrast, we also provide genetic evidence for a growth-promoting, rather than inhibiting, function of *Cdkn1c* in the developing cerebral cortex (Fig. 7c). Two possible scenarios

could explain the seemingly opposite functions inferred from previous and our genetic loss of function analysis. First, the tissue-specific *Cdkn1c* growth-promoting function observed in the developing cortex (this study) could not be observed in full knockout[10,20] because the p57[KIP2]-mediated growth inhibition acts in a dominant manner on the whole organism. Thus additional cell-type and/or tissue-specific *Cdkn1c* functions, such as the growth promotion (i.e. prevention of cell death) as revealed by our conditional *Cdkn1c* analysis, may not unfold. Alternatively, if we consider the architecture of the genomic *Cdkn1c* locus and deletion analysis comparatively, a small region of the 3′ genomic *Cdkn1c* locus could be essential for nascent projection neuron survival and cortical growth (Supplementary Fig. 11). The critical element in the genomic *Cdkn1c* locus can be narrowed down to approximately 500 bp by comparing the deletion[10] leading to full *Cdkn1c* knockout and the conditional deletion of the allele analyzed in this study[24] (Supplementary Fig. 11). Importantly, since *Cdkn1c* shows maternal specific expression, the silent paternal allele was, so far, considered nonfunctional. In stark contrast to this view, we have observed striking phenotypes upon maternal and paternal *Cdkn1c* deletion in cortical progenitors and neurons. A detailed analysis of allelic expression of *Cdkn1c*, using SNP- and UPD-based approaches, however detects strong maternal expression with minimal contribution from the paternal allele. Although we cannot formally rule out a function of the minimal paternal expression, our data indicate that the *Cdkn1c* locus carries the critical functional element which albeit is not under the control of genomic imprinting. While the global growth-inhibiting function of p57[KIP2] protein may or may not be disrupted in the cKO-*Cdkn1c*-MADM-7, the loss of the growth-promoting function inherent to the 500 bp element likely acts in a dominant manner in the conditional- and single-cell MADM deletion paradigms. This interpretation is supported by the fact that *Cdkn1c* full knockout[10] (where the 500 bp element is preserved) do not show any signs of apoptosis in the developing cortex and exhibit macrocephaly[14], likely as a result of non-cell-autonomous mechanisms due to global loss of p57[KIP2]. The here identified cell-autonomous growth-promoting function of genomic *Cdkn1c* locus is highly dosage sensitive with heterozygous cells showing haploinsufficiency. Strikingly, the prevention of apoptosis by intact *Cdkn1c* locus is not subject to genomic imprinting, as mentioned above, in contrast to the growth-inhibitory p57[KIP2] function. In the future it will be revealing to dissect the interplay of endogenous p57[KIP2] protein and the here identified genetic element in the regulation of cortical stem cell behavior. Interestingly, it has been shown that the intracellular distribution and localization of p57[KIP2] plays instructive roles in proliferation and differentiation of adult skeletal muscle stem cells[31]. Whether similar mechanisms and/or in combination with the 500 bp element are relevant for RGP proliferation, cortical neuron production and/or cellular survival represent critical open questions that should be clarified in further studies. Regardless of the precise mechanism, it will be also important to determine the cell-autonomous and non-autonomous downstream signaling components of the *Cdkn1c* locus which promote projection

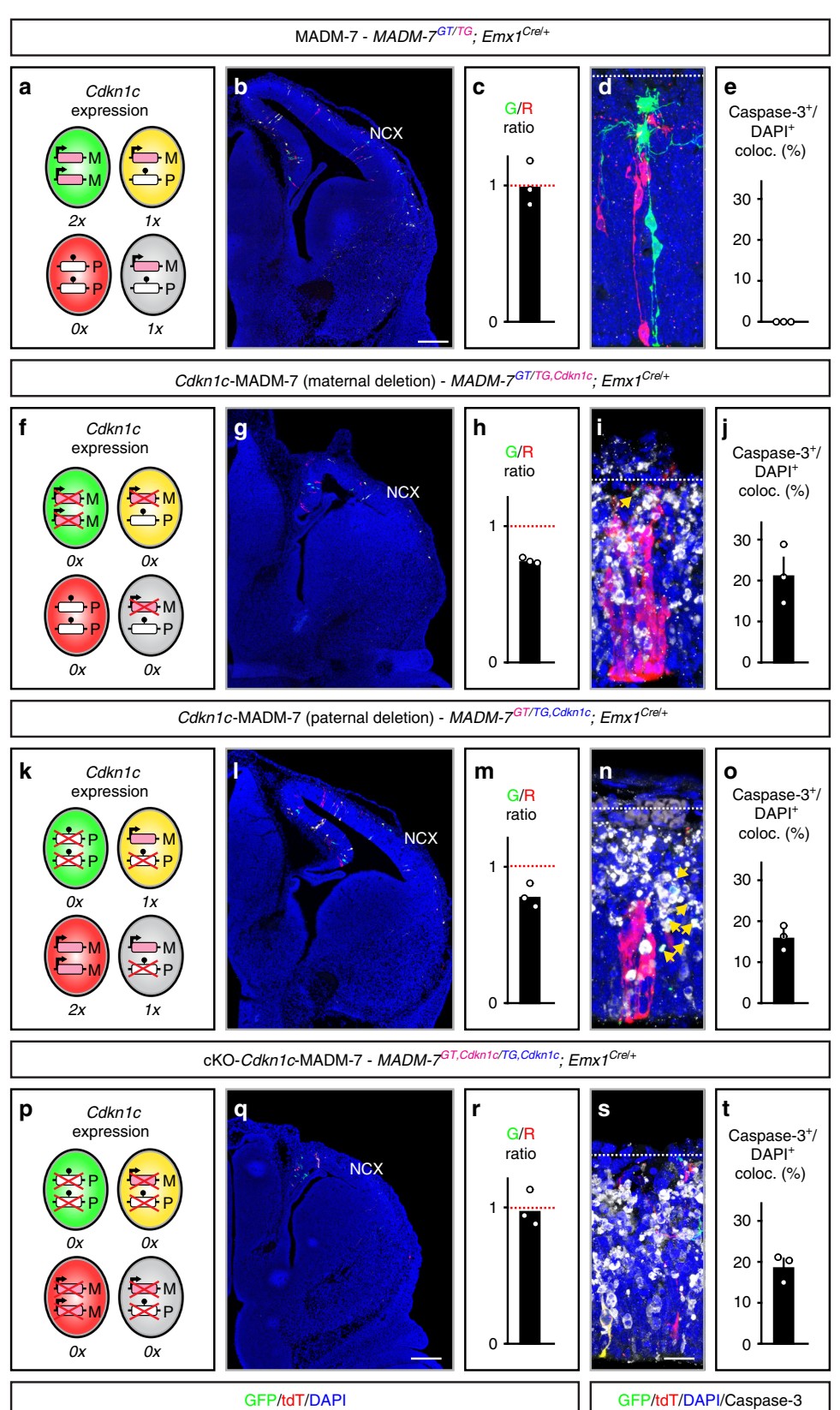

neuron survival and/or regulate global growth of the developing cerebral cortex. Finally, it is intriguing to note that while the intron−exon boundaries in the *Cdkn1c* genomic locus appear to be highly conserved from mouse to human, the exon sequences have diverged considerably in humans[32,33]. It will thus be revealing to determine the generality of our findings in other species than the mouse and in particular in human context. More generally, our study highlights the importance to probe the

**Fig. 5 Cdkn1c function is required for nascent cortical projection neuron survival.** Analysis of developing cerebral cortex in MADM-7 (MADM-7$^{GT/TG}$; Emx1$^{Cre/+}$) (**a–e**), Cdkn1c-MADM-7 (MADM-7$^{GT/TG,Cdkn1c}$;Emx1$^{Cre/+}$) with maternal deletion (**f–j**), Cdkn1c-MADM-7 (MADM-7$^{GT/TG,Cdkn1c}$;Emx1$^{Cre/+}$) with paternal deletion (**k–o**), and cKO-Cdkn1c-MADM-7 (MADM-7$^{GT,Cdkn1c/TG,Cdkn1c}$;Emx1$^{Cre/+}$) (**p–t**) at E13. The parent, from which the MADM cassettes and with recombined Cdkn1c-flox allele (Cdkn1c) (**f–t**) was inherited, is indicated in the respective genotypes in pink (mother) and blue (father). Schematics (**a, f, k, p**) depict green (GFP$^+$), red (tdT$^+$), yellow (GFP$^+$/tdT$^+$) and unlabeled MADM cells with UPD (red, green) and control cells (yellow, unlabeled). Imprinting (arrow, expression; ball on stick, repression) and expression (0×, 1×, 2×) status is indicated. Conditional deletion of Cdkn1c is marked with red cross. Parental origin of chromosome is indicated (M maternal, P paternal). MADM-labeling in overview (GFP$^+$, green; tdT$^+$, red; GFP$^+$/tdT$^+$, yellow) (**b, g, l, q**) and G/R ratio (geometric mean) of single MADM-labeled cortical projection neurons (**c, h, m, r**) in MADM-7 (**b, c**), Cdkn1c-MADM-7 with maternal deletion (**g, h**), Cdkn1c-MADM-7 with paternal deletion (**l, m**), and cKO-Cdkn1c-MADM-7 (**q, r**) at E13 indicate emerging microcephaly. Note the equipotency of cells with matUPD and patUPD in (**c**) and (**r**) but decreased numbers of Cdkn1c$^{-/-}$ cells when compared to Cdkn1c$^{+/+}$ cells in (**h**) and (**m**) regardless of the UPD status. Labeling of apoptotic Caspase-3$^+$ cells (white) in MADM tissue (GFP$^+$, green; tdT$^+$, red; GFP$^+$/tdT$^+$, yellow) (**d, i, n, s**); and quantification of Caspase-3$^+$/DAPI$^+$ coloc (%) (**e, j, o, t**) in MADM-7 (**d, e**), Cdkn1c-MADM-7 with maternal deletion (**i, j**), Cdkn1c-MADM-7 with paternal deletion (**n, o**), and cKO-Cdkn1c-MADM-7 (**s, t**) at E13. Note that in MADM-7 almost no Caspase-3$^+$ cells are detected. Yellow arrows indicate GFP$^+$ remnants of Cdkn1c$^{-/-}$ mutant cells. All bars indicate mean. Error bars represent SEM (**e, j, o, t**). Data points indicate individual animals (n = 3). Nuclei were stained using DAPI (blue). NCX neocortex. Scale bar, 200 μm (**b, g, l, q**) and 20 μm (**d, i, n, s**). Source data are provided as a Source Data file.

relative contributions of the cell intrinsic gene function and extrinsic tissue-wide mechanisms to the overall phenotype not only in health but also in neurodevelopmental disease conditions.

## Methods

**Mouse lines and maintenance.** Mouse protocols were reviewed by institutional ethics committee and preclinical core facility (PCF) at IST Austria and all breeding and experimentation was performed under a license approved by the Austrian Federal Ministry of Science and Research in accordance with the Austrian and EU animal laws. Mice were maintained and housed in animal facilities with a 12-h day/night cycle and adequate food/water conditions according to IST Austria institutional regulations. Mouse lines with Chr. 7 MADM cassettes[18] (MADM-7-GT JAX stock # 021457, MADM-7-TG JAX stock # 021458), Cdkn1c-flox allele[24], Emx1-Cre[34] (JAX stock # 005628) have been described previously. Calculation of recombination probability and recombination of Cdkn1c-flox allele onto chromosomes carrying the MADM cassettes was performed according to standard techniques. These are described in detail elsewhere[17,35] and are summarized in Supplementary Fig. 1. All MADM-based analyses were carried out in a mixed C57BL/6J, CD1 genetic background, in male and female mice without sorting experimental cohorts according to sex. Based on genotype experimental groups were randomly assigned. For allelic expression analysis, we used the inbred mouse lines CAST/EiJ (JAX stock # 000928) and C57BL/6J (JAX stock # 000664). Age of experimental animals is indicated in the respective figures, figure legends and source data file.

**Tissue collection and immunohistochemistry.** Mice were deeply anesthetized by injection of a ketamine/xylazine/acepromazine solution (65 mg, 13 mg and 2 mg/kg body weight, respectively) and unresponsiveness was confirmed through pinching the paw. The diaphragm of the mouse was opened from the abdominal side to expose the heart. Cardiac perfusion was performed with ice-cold PBS (phosphate-buffered saline) followed immediately by 4% PFA (paraformaldehyd) prepared in PB buffer (Sigma-Aldrich). Brains were removed and further fixed in 4% PFA O/N to ensure complete fixation. Brains were cryopreserved with 30% sucrose (Sigma-Aldrich) solution in PBS for approximately 48 h. Brains were then embedded in Tissue-Tek O.C.T. (Sakura). Pregnant females were sacrificed at the respective time points to obtain E13 and E16 embryonic brain tissue. Embryonic and P0 brains were directly transferred into ice-cold 4% PFA. Cryopreservation was performed in 30% sucrose in PBS and embedding was performed according to standard techniques.

For adult time points, 45 μm coronal sections were collected in 24 multi-well dishes (Greiner Bio-one) and stored at −20 °C in antifreeze solution (30% v/v ethyleneglycol, 30% v/v glycerol, 10% v/v 0.244 M PO$_4$ buffer) until used. Adult brain sections were mounted onto superfrost glass-slides (Thermo Fisher Scientific), followed by three wash steps (5 min) with PBS. Tissue sections were blocked for 30 min in a buffer solution containing 5% normal donkey serum (Thermo Fisher Scientific), 1% Trition X-100 in PBS. Primary antibodies in blocking buffer were incubated overnight at 4 °C. Sections were washed three times for 5 min each with PBT (1% Triton X-100 in PBS) and incubated with corresponding secondary antibody diluted in PBT for 1 h. Sections were washed two times with PBT and once with PBS. Nuclear staining of brain sections was performed by 10 min incubation with PBS containing 2.5% DAPI (4',6-diamidino-2-phenylindole, Thermo Fisher Scientific). Sections were embedded in mounting medium containing 1,4-diazabicyclooctane (DABCO; Roth) and Mowiol 4-88 (Roth) and stored at 4 °C until use. Embryonic brains and early postnatal brains (P0) were sectioned at 20 or 30 μm and directly mounted onto superfrost glass-slides (Thermo Fisher Scientific). Immunohistochemistry for embryonic brains was performed as described above. The M.O.M. Immunodetection Kit (Vector

Laboratories) was used by following the instructions of the user manual whenever mouse primary antibody was needed for staining. For a detailed list of antibodies and respective dilutions, vendors and order numbers, see Supplementary Table 1.

**EdU labeling experiments.** Cell-cycle experiments were based on the use of the Click-iT Alexa Fluor 647 imaging kit (Thermo Fisher C10340). Reagents were reconstituted according to the user manual. Pregnant females were injected with EdU (1 mg/ml EdU stock solution; 50 μl/10 g mouse) at E13. Embryos were collected 1 h after EdU injection and 24 h after EdU injection. Tissue was fixed in 4% PFA and immunohistochemistry was performed as described above, except that the Click-iT imaging kit was used (according to the instruction manual) to visualize the EdU signal before performing the DAPI staining.

**Imaging and analysis of MADM-labeled brains.** Sections were imaged using an inverted LSM800 confocal microscope (Zeiss) and processed using Zeiss Zen Blue software. Confocal images were analyzed in Photoshop software (Adobe) by manually counting MADM-labeled cells based on respective marker expression. To determine cortical thickness solely DAPI images were used. Images were opened in Zen Blue software and measurements were performed using the "line"-tool of this software. Three different measurements in the somatosensory cortex were done per image and accordingly combined to one value by averaging the three measured parameters. Six replicates were measured per individual brain. Three different individuals were analyzed per genotype. Significance was determined using two-tailed t test in Graphpad Prism 7.0.

**Single-cell suspension and FACS of MADM-labeled cells.** Pregnant females were sacrificed and E13/E16 embryos were collected. Neocortex area was dissected. Single individuals were used as replicates. Single-cell suspensions were prepared by using Papain containing L-cysteine and EDTA (vial 2, Worthington), DNase I (vial 3, Worthington), Ovomucoid protease inhibitor (vial 4, Worthington), EBSS (Thermo Fisher Scientific), DMEM/F12 (Thermo Fisher Scientific), FBS and HS. All vials from Worthington kit were reconstituted according to the manufacturer's instructions using EBSS. The dissected brain area was directly placed into Papain-DNase solution (20 units/ml papain and 1000 units DNase). Enzymatic digestion was performed for 30 min at 37 °C in a shaking water bath. Next, solution 2 (EBSS containing 0.67 mg Ovomucoid protease inhibitor and 166.7 U/ml DNase I) was added, the whole suspension was thoroughly mixed and centrifuged for 5 min at 1000 rpm at RT. Supernatant was removed and cell pellet was resuspended in solution 2. Trituration with p1000 pipette was performed to mechanically dissolve any remaining tissue parts. DMEM/F12 was added to the cell suspension as a washing solution, followed by a centrifugation step of 5 min with 1500 rpm at RT. Cells were resuspended in media (DMEM/F12 containing 10% FBS and 10% HS) and kept on ice until sorted. Right before sorting, cell suspension was filtered using a 40 μm cell strainer. FACS was performed on a BD FACS Aria III using 100 nozzle and keeping sample and collection devices (0.8 ml PCR tubes) at 4 °C. Duplet exclusion was performed to ensure sorting of true single cells. Cells were sorted into 4 μl lysis buffer (0.2% Triton X-100, 2U/μl RNase Inhibitor (Clonetech)). Immediately after sorting was completed samples were transferred into a 96-well plate (Bio-Rad) that was kept on dry ice. Once the plate was full it was sealed and kept at −80 °C until further processing. For relative Cdkn1c expression experiments GFP$^+$, tdT$^+$ and GFP$^+$/tdT$^+$ cells (200−400 cells) were collected. For the WT-pool and cKO-pool, all MADM-labeled cells (GFP$^+$, tdT$^+$ and GFP$^+$/tdT$^+$) were sorted together in order to isolate only Emx1$^+$ lineages (which express Cre and thus show deleted Cdkn1c locus), ensuring sufficient number of cells (400 cells) as starting material. Sequencing libraries were prepared following the Smart-Seq v2 protocol[36] using custom reagents (VBCF GmbH) and libraries from a 96-well plate were pooled, diluted and sequenced on a HiSeq 2500 (Illumina) using v4 chemistry or NextSeq550 (Illumina).

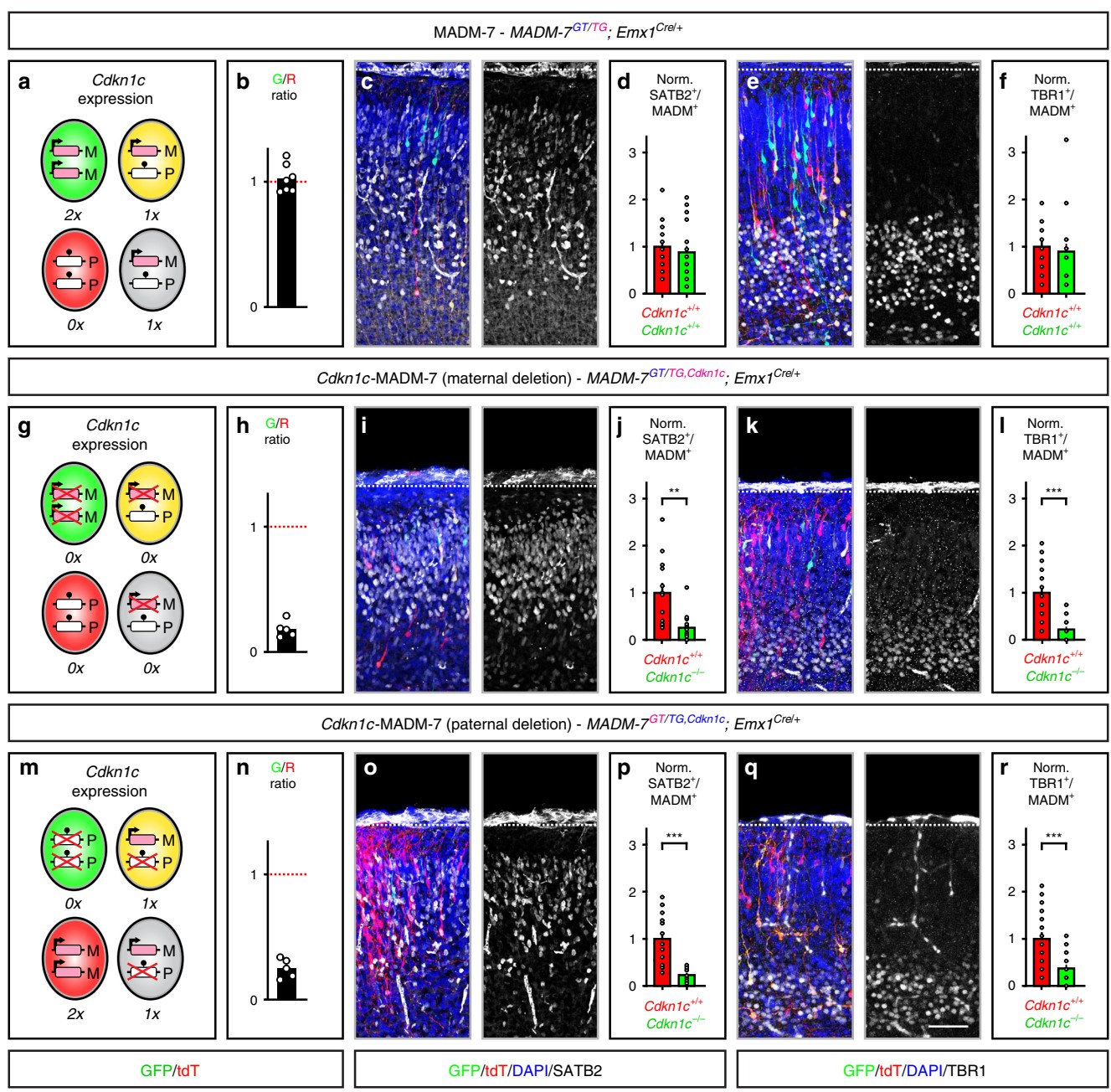

**Fig. 6 Callosal and corticofugal projection neuron classes require *Cdkn1c* for survival.** Analysis of cortical layer marker expression (SATB2 and TBR1) in MADM-7 (*MADM-7^GT/TG;Emx1^Cre/+*) (**a–f**), Cdkn1c-MADM-7 (*MADM-7^GT/TG,Cdkn1c;Emx1^Cre/+*) with maternal deletion (**g–l**), and *Cdkn1c*-MADM-7 (*MADM-7^GT/TG,Cdkn1c;Emx1^Cre/+*) with paternal deletion (**m–r**), in cerebral cortex at P0. The parent, from which the MADM cassette and with recombined *Cdkn1c-flox* allele (*Cdkn1c*) was inherited, is indicated in the respective genotypes in pink (mother) and blue (father). Schematics (**a, g, m**) depict green (GFP+), red (tdT+), yellow (GFP+/tdT+) and unlabeled MADM cells with UPD (red, green) and control cells (yellow, unlabeled). Imprinting (arrow, expression; ball on stick, repression) and expression (0×, 1×, 2×) status of *Cdkn1c* is indicated. Conditional deletion of *Cdkn1c* is marked with red cross. Parental origin of chromosome is indicated (M maternal, P paternal). G/R ratio (**b, h, n**) of single MADM-labeled cortical neurons is depicted as geometric mean. Data points indicate individual animals (n = 5–7). Labeling of upper layer callosal SATB2+ cells (white) (**c, i, o**) and lower layer corticofugal TBR1+ cells (white) (**e, k, q**) in MADM tissue (GFP+, green; tdT+, red; GFP+/tdT+, yellow); and quantification of SATB2+/MADM+ (**d, j, p**) and of TBR1+/MADM+ (**f, l, r**) cells. Red cell numbers are centered to 1 and green cell numbers are shown relative to red (**p < 0.01, ***p < 0.001, two-tailed t test). All bars are mean. Error bars represent SEM (**d, j, p** and **f, l, r**). Note the decrease in GFP+ mutant double-positive neurons when compared to tdT+ wild-type cells. Data points indicate individual animals (n = 5–7) for G/R ratio; and individual sections (n = 15–22) for layer marker expression. Nuclei were stained using DAPI (blue). Scale bar, 50 µm. Source data are provided as a Source Data file.

**Single-cell suspension and antibody-based FACS.** Single-cell suspension was prepared using Neural Tissue Dissociation Kit (Miltenyi #130-092-628) following the manufacturer's instructions. Pregnant females were sacrificed by cervical dislocation. E12 embryos were isolated and directly transferred into 1× PBS and kept on ice. Region of interest (neocortex) was dissected and used for single-cell suspension preparation. Enrichment for neurons and RGPs (progenitors) was performed according to a protocol described previously[23]. The following antibodies were used for FACS: Prominin-1-PEVio770 1:100 (Miltenyi, Clone: MB9-3G8, #130-102-891), CD15-BV421 (5 µl per sample) (BD, Clone: W6D3, #740086), CD11b-APCR700 1:100 (BD, Clone: m1/70, #564985), CD31-APCR700 1:100 (BD, Clone MEC13.3, #565509) and CD45-APCR700 1:100 (BD, Clone: 30-F11, #565478). In order to avoid contamination of nonneuronal and nonprogenitor cell

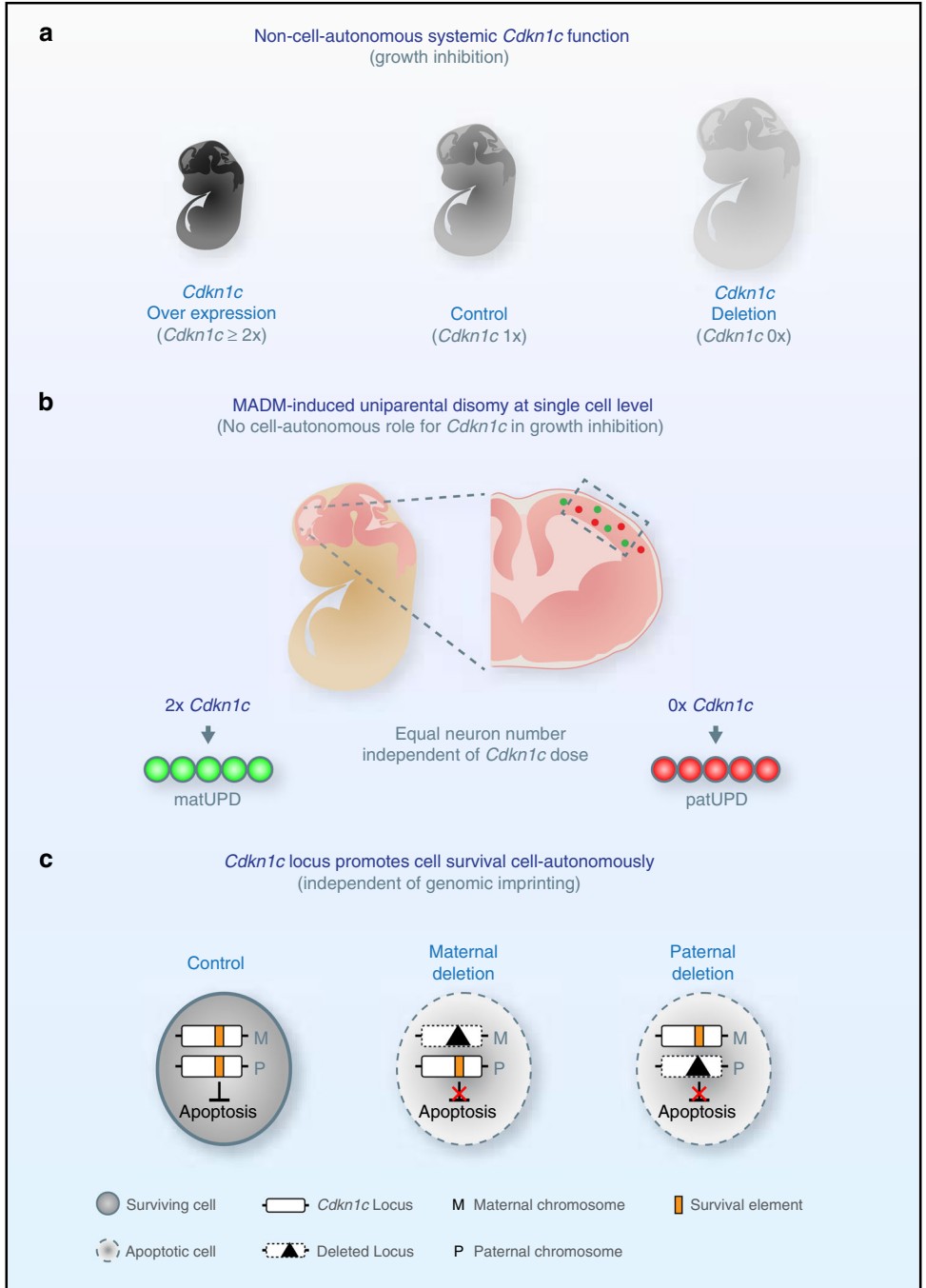

**Fig. 7 Schematic of *Cdkn1c* locus functions in cerebral cortex growth and development. a** Previous studies[41–44] have established that twofold or higher systemic (whole animal) expression of *Cdkn1c* (2×, dark gray) results in embryonic growth inhibition when compared to wild-type embryos with imprinted expression of *Cdkn1c* (1×, gray) whereas global knockout of *Cdkn1c* (0×, light gray) results in embryonic overgrowth[14,45]. **b** Analysis of single cells with MADM-induced uniparental chromosome disomy (UPD) (twofold overexpression (green cells) or no expression of *Cdkn1c* (red cells)) in an otherwise wild-type environment indicates no cell-autonomous function for *Cdkn1c* in cortical growth regulation. **c** Conditional *Cdkn1c* locus deletion and analysis at single-cell resolution revealed a putative survival element (SE) acting in a dosage-sensitive manner but which is not subject to genomic imprinting. Deletion of either the maternal or paternal *Cdkn1c* locus leads to p53-mediated apoptosis.

populations an initial gating was performed where cells positive for CD11b, CD31 and CD45 were eliminated. The cell population negative for these markers was then used further. The progenitor cell population which was Prominin[+]/CD15[+] was collected. The neuronal population was negative for both markers. Cell sorting was performed at 4 °C using the 100 nozzle and on an FACS Aria III, BD Biosciences. Cells were collected in a custom-made RNA extraction buffer (30 nM TRIS pH 8, 10 nM EDTA pH 8, 1% SDS and 200 µg/µl Proteinase K).

**RNA extraction.** Upon cell sorting, samples were incubated for 30 min at 37 °C. Total volume was adjusted to 250 µl using RNase-free $H_2O$ (Thermo Fisher

Scientific) followed by addition of 750 µl Trizol LS (Thermo Fisher Scientific). Samples were mixed by inverting five times. After a 5-min incubation step at RT, the entire solution was transferred into a MaXtract tube (Qiagen). Two hundred microliters chloroform (Sigma-Aldrich) was added, followed by three times 5-s vortexing and 2-min incubation at RT. Samples were centrifuged for 2 min with 12,000 rpm at 18 °C. Supernatant was transferred to a new tube and isopropanol (Sigma-Aldrich) was added in a 1:1 ratio. For better visibility of the RNA pellet, 1 µl GlycoBlue (Thermo Fisher Scientific) was added and entire solution was mixed by vortexing (3 × 5 s). Samples were left for precipitation o/n at −20 °C. After precipitation samples were centrifuged for 20 min with 14,000 rpm at 4 °C.

Supernatant was removed and RNA pellet was washed with 70% ethanol, followed by a 5-min centrifugation step (14,000 rpm at 4 °C). RNA pellet was resuspended in 12.5 µl RNase-free H$_2$O. RNA quality was analyzed using Bioanalyzer RNA 6000 Pico kit (Agilent) following the manufacturer's instructions. RNA samples were stored at −80 °C until further use.

**Analysis of allelic expression by Sanger and deep sequencing**. RNA was isolated from 10,000 cells (sorted as described above) using Trizol LS (Thermo Fisher Scientific), DNAse1 treated (DNA free KIT, Ambion) and converted to cDNA (SuperScript VILO, Invitrogen). Genomic DNA was prepared from the carcass of one embryo used for FACS sorting. For Sanger sequencing, regions of interest were amplified (Platinum Hot Start PCR Master Mix, Invitrogen) and sequenced using a nested primer (LGC Genomics GmbH, primers given below). Chromatograms were prepared using partly customized functions from R package sangerseqR[37]. For deep sequencing, regions of interest were amplified with different primers that produce fragments where the SNP of interest is located within 50 bp of one end, with a high-fidelity DNA polymerase (Q5 NEB, primer given below). PCR fragments were purified (Ampure XP, Beckman Coulter) and pooled per sample of origin. Each pool was prepared for deep sequencing (VBCF GmbH, Vienna) and sequenced on a HiSeq 2500 (Illumina) using v4 chemistry. Each sequencing produced a minimum of 700,000 uniquely aligned reads (>99% of total reads), using STAR v.2.5.0c[38] and mouse genome build GRCm38. Samtools v1.3 was used to prepare pileups and the occurrences of high-quality nucleotides were counted at two positions (Cdkn1c: 143458858, Ndn: 62348457, GRCm38) using custom scripts and the relative abundance of each nucleotide at the SNP position plotted. Details on primers for Sanger sequencing: Cdkn1c (TC SNP), PCR: GGGACTTCAACTT CCAGCAG, CTCAGAGACCGGCTCAGTTC, sequencing: TCTGTGCCCGCCTT CTAC. Cdkn1c (AG SNP), PCR: TAGAGGCTAACGGCCAGAGA, GCTTTACAC CTTGGACCAG, sequencing: CTGGGACCTTTCGTTCATGT. Ndn (AG SNP), PCR: CACTTCCTCTGCTGGTCTCC, TGCTTCTGCACCATTTCTTG, sequencing: TCCTTCACCAACACGTACCA. Details on primers for deep sequencing: Ndn (AG SNP), PCR: CACTTCCTCTGCTGGTCTCC, GCTGTCCTGCATCTCA CAGT. Cdkn1c (AG SNP), PCR: CCACGGTTTTGTGGAAATCT, GTGGGGGCT TTTACTCAACA. Note that the Ndn SNP is shown on the reverse strand.

**Statistical analysis of RNA-seq data**. Raw reads were delivered as unaligned BAM files and technical replicates were combined using samtools (v1.8). For alignment, BAM files were converted to fastq format using bamToFastq (bedtools suite v2.26.0). Reads within transcripts were counted using STAR (v2.5.0c) with an index prepared from GRCm38 and Gencode M16 gene annotation. Parameters used for STAR alignment: outFilterMultimapNmax 1, outSAMstrandField intronMotif, outFilterIntronMotifs RemoveNoncanonical and quantMode GeneCounts. All downstream analyses were performed in R (v3.4.4) using DESeq2 (v1.16.1).

For Fig. 1 we analyzed 39 samples and removed 7 samples with a low percentage of uniquely aligned reads (<40%) or consistently low correlation with other biological replicates to reduce noise (see Supplementary Data 1 for details on samples used in the analysis). For DESeq2 analyses genes with a mean coverage of less than ten reads, over all samples under investigation, were removed. Figure 1c: size factor normalized read counts (DESeq2) were used. Relative expression values for Cdkn1c gene were calculated by dividing the normalized read counts for each sample by the median of the normalized read counts of the respective control samples. Figure 2b: Relative expression values for indicated genes were calculated by dividing the normalized read counts for each matUPD and patUPD sample by the mean of the normalized read counts of the respective matUPD sample. Significance of differential expression between groups was calculated for each developmental time point using DESeq2 with the following parameters: design = ~sample + group. See Supplementary Data 1 for details on samples. Supplementary Data 2 provides full differential expression results. To identify changes in cell-cycle state between matUPD and patUPD cells, we compiled a list of 78 genes that were shown to indicate cell-cycle stages in RNA-seq data[39]. These genes are also indicated in Supplementary Data 2.

Figure 3: Relative expression of Cdkn1c was calculated as for Fig.1 using the mean normalized read count of control cells to calculate relative expression. Note that for the purpose of this manuscript only Cdkn1c read counts were used and the full dataset together with an in-depth analysis will be published at a later time.

Figure 4: Genes with a mean coverage of less than ten reads over all samples under investigation were removed. Figure 4b: Principal component analysis was performed on variance stabilized count data calculated by varianceStabilizingTransformation (DESeq2) using the top 500 most variable genes and parameter: blind = T. Figure 4c: relative expression of Cdkn1c was calculated and plotted as for Fig.1 using the mean normalized read count of control cells to calculate relative expression. Significance of differential expression between groups was calculated for each developmental time point using DESeq2 with parameter: design = ~group. Supplementary Data 3 provides details on samples and Supplementary Data 4 provides full results of differential expression analysis. Figure 4e: Significance scores were calculated as the log$_{10}$ of the adjusted p value. Prefix of this score was adjusted to represent upregulation (positive) or downregulation (negative). The position of each gene was calculated as the midpoint of its annotated genomic locus. Only informative genes telomeric to position 141459694 (mm10) are shown. The putative maximum size of the Kcnq1 cluster is

defined here as the region spanning from Ascl2 to Ano1 genes (chr7:142966829-144738543, mm10). Figure 4f: Gene Ontology term enrichment was performed using genes with an adjusted p value < 0.05, separately for upregulated genes (log$_2$ fold-change > 0, higher in cKO) or downregulated genes (log$_2$ fold-change < 0, higher in control) using enrichGO (clusterProfiler v3.4.4) with org.Mm.eg.db (v3.4.1) and parameters: keytype = "ENTREZID", ont = "ALL", pool = T, readable = T, pvalueCutoff = 0.05, qvalueCutoff = 0.1, maxGSSize = 2000. As the universe parameter, we used all informative genes used in this analysis. To reduce the number of enriched GO terms for display, we used a combination of manual curation and REVIGO (http://revigo.irb.hr/)[40] with parameters: Allowed similarity: Tiny (0.4), similarity measure: Jiang and Conrath. Details on all enriched GO terms and manual/REVIGO curation are provided in Supplementary Data 5a, b. Supplementary Fig. 3, 4: Read coverages across the genome were calculated using bam2wig.py (RseQC v2.6.3) using parameter: -t 1000000. Resulting wiggle files were converted to BED format using convert2bed (bedops v2.4.14) and regions of interest were extracted using intersectBed (bedtools v2.25.0). Averaging of coverages and final wiggle file preparation was done in R. Wiggle files were displayed on the UCSC genome browser (https://genome.ucsc.edu/). Reference annotation-based transcript (RABT) assembly was performed for each replicate using cufflinks (v2.2.1) with Gencode M16 annotation using standard parameters. Merging of RABT assemblies was done using cuffmerge (v2.2.1) using standard parameters. No novel transcripts were identified in the Cdkn1c locus.

**Reporting summary**. Further information on research design is available in the Nature Research Reporting Summary linked to this article.

## Data availability
Data that support the findings of this study have been deposited at the NCBI GEO data repository under accession number GSE138230. The source data underlying Figs. 1c, f, 2b, d, f, 3b, h, n, e, k q, f, l, r, 4c, 5c, h, m, r, e, j, o, t, 6b, h, n, d, j, p, f, l, r, Supplementary Figs. 5o, p, q, 6a, 7b, d, 8a-d, 9d, g, j and 10c, f, i, l are provided as a Source Data file.

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

## Acknowledgements

We thank A. Heger (IST Austria Preclinical Facility), A. Sommer and C. Czepe (VBCF GmbH, NGS Unit) and C. Streicher for mouse colony maintenance, assistance, and/or technical help with RNA-seq; Y. Gotoh and all members of the Hippenmeyer lab for discussion. N.A received support from Austrian Science Fund (FWF) Firnberg-Programm (T 1031). R.B. received support from Austrian Science Fund (FWF) Meitner-Programm (M 2416). This work was also supported by IST Austria institutional funds; a grant from NÖ Forschung und Bildung n[f + b] (C13-002) to S.H.; and the European Research Council (ERC) under the European Union's Horizon 2020 research and innovation program (grant agreement no. 725780 LinPro) to S.H.

## Author contributions

S.L., R.B., F.M.P. and S.H. conceived the research. S.L., R.B., F.M.P. and S.H. designed all experiments and interpreted the data. S.L., R.B., F.M.P. and N.A. performed all the experiments. K.I.N. provided reagents. S.H. wrote the manuscript with inputs from S.L., R.B. and F.M.P. All authors edited and proofread the manuscript.

## Competing interests

The authors declare no competing interests.
