## [Peer Review File · Nature Communications]

Reviewers' Comments:

Reviewer #1:

Remarks to the Author:

The manuscript entitled "Unexpected role of imprinted *Cdkn1c* genomic locus in cerebral cortex development" by Susanne Laukoter and colleagues reports a study aimed to investigate how p57 (or its encoding gene, *Cdkn1c*) participates to the generation of cortical projection neurons by radial glia progenitor (RGP). As a matter of facts, some previous literature data suggest that this Cip/Kip protein is required for correct corticogenesis. The conclusion is mostly (but not only) based on alterations of cortical cytoarchitecture subsequent to p57 quantitative alterations. Ablated *Cdkn1c* mouse show macrocephaly and cortical hyperplasia. The picture is in accord with the idea that p57 is mainly a cdk inhibitor and thus, a growth inhibitor factor. However, it now appears that p57 plays a plethora of functions and that most of the mechanisms of its phenotypical effects might not to be correlated to growth inhibition. In addition, from a structural point of view, p57 is almost completely an unstructured protein. The feature increases enormously the protein plasticity, potential targets and the possibility of p57 to be post-synthetically regulated. Importantly, the locus where *Cdkn1c* maps is subject to a complex imprinting process and thus, only the maternal allele is expressed (the paternal is almost completely silenced). Moreover, the maternal allele is also subject to transcriptional control due to different transcriptional factors that make *Cdkn1c* expression active only in specific tissues and in peculiar period of time during development. Thus, also the expression of maternal allele occurs in specific context and is finely controlled.

The authors aimed to clarify the role of p57 in corticogenesis mostly employing genetic strategies that allow the generation of uniparental chromosome disomy (UPD) in sparse RGD cells. In addition, and importantly, these cells might be detected on the basis of markers that allow their clear microscopy identification and quantification.

In brief, few cells showed 2 copies of the maternal allele (and were green coloured) or 2 copies of paternal allele (red) or both the alleles (yellow). The majority of the cells, however, did not expressed tagged alleles and resulted unlabeled. When the authors evaluated the content of green and red cells, they found a ratio of about 1. This is in contrast with the idea that p57 is a major regulator of proliferation, since the growth of green cells should be minor than that of red cells. In my view, however, the result is not excessively unexpected, since the authors did not analyze the level of p57 in cells where the two maternal alleles are present and do not clarify p57 interaction, localization and functions. Moreover, the complexity of the mechanisms regulating the growth might exceed the importance of p57 content. Finally, the putative (i.e. not linked to growth control) p57 roles might indirectly favor the growth when two maternal active alleles are present. In any case, the authors concluded that "observed macrocephaly in *Cdkn1c*^{-/-} full knockout likely reflects global organism overgrowth". This is possible, but quite unspecific.

Subsequently, a specific gene silencing strategy was employed. Particularly, both cells with disomy of maternal ablated allele or cells with disomy of paternal ablated allele were obtained. Under these conditions, the number of ablated cells (both if the allele deleted is of maternal or paternal origin) is very scarce. Moreover, the modified animals showed microcephaly. These results are not in accord with a growth restraining activity of p57 and allow the conclusion that of "a growth promoting *Cdkn1c* function". This unprecedented discovered function is mainly acting to promote the survival of differentiating and maturing cortical projection neurons rather than in proliferating RGP. This explanation did not regard p57, level but probably a not clarified activity of a small region of *Cdkn1c* gene. Particularly, the authors suggested that a "small region of the 3' genomic *Cdkn1c* locus could be essential for nascent projection neuron survival and cortical growth".

The study is interesting, but it appears too preliminary for being published as it is. Experiments for identifying and functionally characterize the putatively identified *Cdkn1c* region should be made. For example, its capability to be translated or how to increase the growth, or resistance to apoptosis (or other activities, like modulation of other genes). Moreover, studies should clarify the

phenotypical effect of having two copies of maternal *Cdkn1c*, including a full characterization of cell cycle engine as well as effects on the level (if it exists) of p57. It is to stress that p57 seems to control resistance to stress that could result in a different surviving under peculiar conditions and might facilitate growth.

In brief, although the study opens new perspectives, in our view, the data reported are not sufficient to support the promising conclusions.

Reviewer #2:

Remarks to the Author:

Laukoter et al. describe work testing for a cell-autonomous role of the imprinted *Cdkn1c* gene in neural progenitors and early neurons during embryonic mouse corticogenesis. The study was mediated by previous paradoxical findings that constitutive ablation of *Cdkn1c* drives brain overgrowth, while brain specific nestin-CRE mediated ablation drives cortical thinning. The authors use a powerful approach that enables sparse labeling of genetically-modified progenitors and neurons (MADM) to test the impact of both maternal/paternal uniparental disomy and conditional ablation of *Cdkn1c* on proliferation, survival, and cortical thickness.

As *Cdkn1c* exhibits silencing of the paternal allele, the prediction was that uniparental disomy of the paternal allele would result in changes in proliferation. Contrary to expectations, however, the authors found no impact of uniparental disomy on cell autonomous proliferation, despite showing that imprinting was intact in the relevant cells. Next the authors show via MADM sparse targeting that both constitutive and conditional deletion of either the maternal or paternal allele indeed resulted in cell autonomous decreased proliferative output impacting thickness and lamination and that this is due to increased apoptosis.

Finally, the authors hypothesize that the differences between expected imprinting-associated loss of expression and genetic ablation-based loss of expression may be due to a 500bp region (identified from comparison of the "full" and conditional *Cdkn1c* alleles) that is not subject to imprinting. Overall, this study is an excellent example of the need to examine cell autonomous versus organismal effects of specific genes, here *Cdkn1c*. The experimental logic with regard to use of uniparental disomy and MADM is excellent, the results appear to meet expectations of rigor, and the finding is of sufficient interest to warrant publication. One major weakness that must be addressed is that there is insufficient evidence that the experimental model used for studying *Cdkn1c* is valid, i.e. that loss-of-expression occurs and is specific to *Cdkn1c/p57KIP2* in both the paternal uniparental disomy and genetic ablation conditions.

Major points:

1. While the authors show that *Cdkn1c* imprinting is intact in the relevant cells, they do not show that imprinting-mediated loss of expression indeed occurs in their experimental model (i.e. in the MADM-associated paternal uniparental disomy cells).
2. Similarly, considering the unexpected results, the authors should show that it is *Cdkn1c* and not neighboring genes that are specifically impacted by the floxed deletion. This could be done globally via RNA-seq (preferred considering possible long-distance regulatory interactions) or locally via qPCR of nearby genes. Obviously, genes other than *Cdkn1c* could be secondarily impacted, which would be anticipated and in support of direct impact on *Cdkn1c* as long as *Cdkn1c* itself is altered as well.

Minor points:

1. Considering the differences between alleles, it would be useful to show the full gene/locus

genomic structure for the loss-of-function allele in addition to the simple schematics shown in extended fig 1. This could be a supplemental figure.

General Statements:

We sincerely thank the two reviewers for their constructive critiques and thoughtful suggestions to improve our study. We are pleased that the reviewers in principle compliment the high technical quality and rigor of our work; and agree in general with the conclusions of our study. For the revised manuscript, we have added a substantial amount of new data that we believe advances the study. We have rewritten large portions of the manuscript (marked in yellow throughout the text file), incorporated all suggestions of the reviewers and made every effort to streamline the manuscript. The revised manuscript should now provide much more depth and coherence. Below, we first summarize the major new experiments that we have added to this revised manuscript. We then provide a point-by-point response to the specific critiques raised by the reviewers.

Summary of major new experiments and figures:

In our revision we have made every effort to address all the constructive reviewers' critiques by adding new experimental data and/or deepening the analysis. More specifically we,

1. Determined relative *Cdkn1c* expression in MADM-induced UPD by RNA sequencing to validate MADM-induced UPD paradigm for the study of imprinted *Cdkn1c* (incorporated into Figure 1)
2. Assessed any change in expression of a set of cell-cycle regulator genes, and measured cell-cycle properties in MADM-induced UPD (incorporated into Figure 1 and Supplementary Table 2).
3. Determined relative *Cdkn1c* expression in MADM-labelled cells with UPD and upon genetic *Cdkn1c* deletion (i.e. validation of genetic paradigms) (added to Figure 2).
4. Analyzed global gene expression changes upon genetic *Cdkn1c* deletion in cKO by RNA sequencing (new Figure 3).
5. Analyzed haploinsufficiency of genomic *Cdkn1c* locus (new Supplementary Figure 5).
6. Deepened the analysis of cell death upon genetic *Cdkn1c* deletion and determined cell-type (RGPs, nascent neurons) specificity of apoptosis, and reduced level of proliferating RGPs (new Supplementary Figures 6-8).
7. Summarized the main findings of our study in schematics (new Figure 6 and new Supplementary Figure 9).

Below we address the more specific concerns of the reviewers and provide point-to-point responses)

Point-to-Point Response (original reviewer comments are copied in blue):

Reviewer #1: The manuscript entitled “Unexpected role of imprinted *Cdkn1c* genomic locus in cerebral cortex development” by Susanne Laukoter and colleagues reports a study aimed to investigate how p57 (or its encoding gene, *Cdkn1c*) participates to the generation of cortical projection neurons by radial glia progenitor (RGP). As a matter of facts, some previous literature data suggest that this Cip/Kip protein is required for correct corticogenesis. The conclusion is mostly (but not only) based on alterations of cortical cytoarchitecture subsequent to p57 quantitative alterations. Ablated *Cdkn1c* mouse show macrocephaly and cortical hyperplasia. The picture is in accord with the idea that p57 is mainly a cdk inhibitor and thus, a growth inhibitor factor. However, it now appears that p57 plays a plethora of functions and that most of the mechanisms of its phenotypical effects might not to be correlated to growth inhibition. In addition, from a structural point of view, p57 is almost completely an unstructured protein. The feature increases enormously the protein plasticity, potential targets and the possibility of p57 to be post-synthetically regulated. Importantly, the locus where *Cdkn1c* maps is subject to a complex imprinting process and thus, only the maternal allele is expressed (the paternal is almost completely silenced). Moreover, the maternal allele is also subject to transcriptional control due to different transcriptional factors that make *Cdkn1c* expression active only in specific tissues and in peculiar period of time during development. Thus, also the expression of maternal allele occurs in specific context and is finely controlled.

The authors aimed to clarify the role of p57 in corticogenesis mostly employing genetic strategies that allow the generation of uniparental chromosome disomy (UPD) in sparse RGD cells. In addition, and importantly, these cells might be detected on the basis of markers that allow their clear microscopy identification and quantification.

In brief, few cells showed 2 copies of the maternal allele (and were green coloured) or 2 copies of paternal allele (red) or both the alleles (yellow). The majority of the cells, however, did not expressed tagged alleles and resulted unlabeled. When the authors evaluated the content of green and red cells, they found a ratio of about 1. This is in contrast with the idea that p57 is a major regulator of proliferation, since the growth of green cells should be minor than that of red cells.

In my view, however, the result is not excessively unexpected, since the authors did not analyze the level of p57 in cells where the two maternal alleles are present and do not clarify p57 interaction, localization and functions.

We completely agree with the reviewer and have now quantified the relative levels of *Cdkn1c* expression in MADM-induced UPD at E13 and E16 by RNA sequencing. These data validated the experimental MADM paradigm since patUPD expressed very low to nearly undetectable levels of *Cdkn1c* whereas matUPD showed indeed approximately 2-fold higher relative *Cdkn1c* levels when compared to control cells. These data have been incorporated into Figure 1c.

Regarding the second point we absolutely agree that getting more insight into p57 interaction, localization and function is imperative. Due to potential cell-type specific functions of p57 such

analysis will be only informative when performed with single cell resolution and *in vivo*. To this end we first tested the commercially available antibodies described previously (Furutachi et al. 2015; Mademtoglou et al. 2018). However, while these antibodies appear to work very well in cell culture paradigms and/or peripheral tissues we could unfortunately not obtain specific signal in the developing neocortex on cryosections of embryonic tissue. Thus due to the lack of appropriate reagents we are currently not in a position to investigate the above points further. Nevertheless we expect that development of improved assays and reagents will allow us to address these interesting aspects in the future.

Moreover, the complexity of the mechanisms regulating the growth might exceed the importance of p57 content.

We agree with the reviewer and believe that the mechanism regulating the growth may extend beyond *Cdkn1c* function. To address this issue we used data from the RNA sequencing experiments and specifically analyzed expression profiles of cell cycle genes that might be altered due to UPD. The data is presented in Figure 1g and Supplementary Table 2 and shows that none of the analyzed cell cycle genes (2 representative genes are shown in Figure 1g; 72 genes at E13, and 68 genes at E16 were investigated in total) displayed differential expression upon MADM-induced UPD. These data are in agreement with our interpretation that non-cell-autonomous mechanisms at the systemic/tissue level are critical for organismic growth regulation.

Finally, the putative (i.e. not linked to growth control) p57 roles might indirectly favor the growth when two maternal active alleles are present.

The reviewer points out an important aspect and we completely agree that p57 functions (other than growth control) may act synergistically together with any feature or overall cell state upon induction of UPD. Thus far we have however not observed significant cell-autonomous growth phenotypes in cells with matUPD (when compared to patUPD or control). Furthermore, upon conditional genetic *Cdkn1c* deletion we observed growth (i.e. apoptosis) phenotypes that are dominant over the UPD status. It will be intriguing in the future to dissect the specific p57 protein-mediated functions and compare to the functional requirement of intact *Cdkn1c* genomic locus. We now discuss these important points in more depth in the revised *Discussion* of the manuscript.

In any case, the authors concluded that “observed macrocephaly in *Cdkn1c*^{-/-} full knockout likely reflects global organism overgrowth”. This is possible, but quite unspecific.

We thank the reviewer for the constructive comment and completely agree that currently, based on our single cell analysis, the interpretation of global organism overgrowth phenotype in *Cdkn1c* full knockout (Mairet-Coello et al. 2012; Zhang et al. 1997) reflects a hypothesis. In order to rigorously approach this hypothesis it will however be necessary to generate additional

conditional deletion alleles and new transgenic mice. We hope that the reviewer agrees that such efforts reach beyond the scope of the current manuscript.

Subsequently, a specific gene silencing strategy was employed. Particularly, both cells with disomy of maternal ablated allele or cells with disomy of paternal ablated allele were obtained. Under these conditions, the number of ablated cells (both if the allele deleted is of maternal or paternal origin) is very scarce. Moreover, the modified animals showed microcephaly. These results are not in accord with a growth restraining activity of p57 and allow the conclusion that of “ a growth promoting *Cdkn1c* function”. This unprecedented discovered function is mainly acting to promote the survival of differentiating and maturing cortical projection neurons rather than in proliferating RGP’s”. This explanation did not regard p57, level but probably a not clarified activity of a small region of *Cdkn1c* gene. Particularly, the authors suggested that a “small region of the 3’ genomic *Cdkn1c* locus could be essential for nascent projection neuron survival and cortical growth”.

The study is interesting, but it appears too preliminary for being published as it is. Experiments for identifying and functionally characterize the putatively identified *Cdkn1c* region should be made. For example, its capability to be translated or how to increase the growth, or resistance to apoptosis (or other activities, like modulation of other genes).

We thank the reviewer for the constructive comments and suggestions. In order to rigorously identify and characterize the putative critical *Cdkn1c* genomic region we are conceiving a series of new conditional and deletion alleles for the generation of new transgenic mice. These mice in combination with MADM shall allow further analyses at unprecedented resolution but we hope that the reviewer agrees that such efforts reach beyond the scope of the current manuscript.

In any case, we have now in much more detail investigated the consequences of genetic *Cdkn1c* deletion, and deepened the analysis of the increased cell death phenotype. First, we determined global gene expression changes upon *Cdkn1c* ablation in cortical cells by RNA sequencing. Intriguingly these data clearly showed that gene ontology terms related to neurogenesis (decreased) and to cell death (upregulated) were significantly overrepresented. These data are presented in new Figure 3 and Supplementary Tables 6a-6b.

Next we deepened the analysis of cell death upon genetic deletion of *Cdkn1c*. To this end we first determined whether the *Cdkn1c* genomic locus exhibits dosage sensitivity since heterozygous *Cdkn1c* deletion (Figures 2 and 4) results in severe microcephaly comparable to homozygous *Cdkn1c* deletion. To this end we analyzed whether homozygous *Cdkn1c*^{+/+} wild-type cells may exhibit survival advantage when compared to *Cdkn1c*^{+/-} heterozygous cells. Indeed, much less cells with homozygous intact *Cdkn1c* genomic locus were positive for Caspase-3 than heterozygous cells, regardless of the imprinting status. These data are presented in new Supplementary Figure 5, and demonstrate at the functional level that the genetic deletion of *Cdkn1c* genomic locus results in an increased probability of cell death in a highly dosage dependent manner (i.e. demonstrates haploinsufficiency of *Cdkn1c* locus).

We next determined cell-type specificity of *Cdkn1c* genomic locus function and quantitatively assessed the rate of cell death in ventricular RGP’s and nascent neurons in the CP by immunohistochemistry. We found that both nascent cortical projection neurons and RGP’s

require intact *Cdkn1c* for survival. These data are presented in new Supplementary Figure 6. Consequently the overall number of actively proliferating RGP is reduced from early E13 embryonic stages onward (shown in new Supplementary Fig. 7). Lastly we corroborated our findings by analysis of p53 expression which is upregulated upon stress and marks cells that initiate the apoptotic pathway (data shown in new Supplementary Fig. 8).

Moreover, studies should clarify the phenotypical effect of having two copies of maternal *Cdkn1c*, including a full characterization of cell cycle engine as well as effects on the level (if it exists) of p57.

We agree with the reviewer and thus performed additional experiments to characterize the cell cycle engine and the levels of *Cdkn1c* in cells with matUPD (2 copies of *Cdkn1c*).

First we quantified the relative levels of *Cdkn1c* expression in MADM-induced UPD at E13 and E16 by RNA sequencing. These data validated the experimental MADM paradigm since patUPD expressed very low to nearly undetectable levels of *Cdkn1c* whereas matUPD showed indeed approximately 2-fold higher relative *Cdkn1c* levels when compared to control cells. These data have been incorporated into Figure 1.

Next we used expression data from the RNA sequencing experiments and specifically analyzed cell cycle genes that might be altered due to UPD. The data (presented in Figure 1g and Supplementary Table 2 shows that none of the analyzed cell cycle genes (2 representative genes are shown in Figure 1g; 72 genes at E13, and 68 genes at E16 were investigated in total) displayed differential expression upon MADM-induced UPD.

Lastly we directly measured proliferation properties in cortical cells with MADM-induced UPD. To this end we performed EdU pulse-chase experiments. The results are illustrated in Figures 1h-k and demonstrate no significant differences in cells with matUPD (and patUPD) when compared to control.

It is to stress that p57 seems to control resistance to stress that could result in a different surviving under peculiar conditions and might facilitate growth.

We agree with the reviewer and hope that we have sufficiently addressed this point in the above elaboration of our deepened analysis of cell death upon genetic deletion of *Cdkn1c*.

Reviewer #2 (Remarks to the Author):

Laukoter et al. describe work testing for a cell-autonomous role of the imprinted *Cdkn1c* gene in neural progenitors and early neurons during embryonic mouse corticogenesis. The study was mediated by previous paradoxical findings that constitutive ablation of *Cdkn1c* drives brain overgrowth, while brain specific nestin-CRE mediated ablation drives cortical thinning. The authors use a powerful approach that enables sparse labeling of genetically-modified progenitors and neurons (MADM) to test the impact of both maternal/paternal uniparental disomy and conditional ablation of *Cdkn1c* on proliferation, survival, and cortical thickness.

As *Cdkn1c* exhibits silencing of the paternal allele, the prediction was that uniparental disomy of the paternal allele would result in changes in proliferation. Contrary to expectations, however, the authors found no impact of uniparental disomy on cell autonomous proliferation, despite showing that imprinting was intact in the relevant cells. Next the authors show via MADM sparse targeting that both constitutive and conditional deletion of either the maternal or paternal allele indeed resulted in cell autonomous decreased proliferative output impacting thickness and lamination and that this is due to increased apoptosis.

Finally, the authors hypothesize that the differences between expected imprinting-associated loss of expression and genetic ablation-based loss of expression may be due to a 500bp region (identified from comparison of the “full” and conditional *Cdkn1c* alleles) that is not subject to imprinting. Overall, this study is an excellent example of the need to examine cell autonomous versus organismal effects of specific genes, here *Cdkn1c*. The experimental logic with regard to use of uniparental disomy and MADM is excellent, the results appear to meet expectations of rigor, and the finding is of sufficient interest to warrant publication. One major weakness that must be addressed is that there is insufficient evidence that the experimental model used for studying *Cdkn1c* is valid, i.e. that loss-of-expression occurs and is specific to *Cdkn1c/p57KIP2* in both the paternal uniparental disomy and genetic ablation conditions.

We thank the reviewer for encouragement and are pleased that s/he finds the experimental logic excellent, the results rigorous and the findings of interest. We completely agree with the reviewer that more validation of the experimental models (UPD and conditional deletion) is necessary. We made every possible effort to address this critical point (see below *Major points 1-2* for details), besides further investigations and deepening of the analysis (see also above *Summary of major new experiments and figures*).

Major points:

1. While the authors show that *Cdkn1c* imprinting is intact in the relevant cells, they do not show that imprinting-mediated loss of expression indeed occurs in their experimental model (i.e. in the MADM-associated paternal uniparental disomy cells).

We completely agree with the reviewer and have now quantified the relative expression levels of *Cdkn1c* in MADM-induced UPD at E13 and E16 by RNA sequencing. These data validated the experimental MADM paradigm since patUPD expressed very low to nearly undetectable levels of *Cdkn1c* whereas matUPD showed indeed approximately 2-fold higher relative *Cdkn1c* levels when compared to control cells. These data have been incorporated into Figure 1c.

In order to corroborate the above results we have also analyzed the relative *Cdkn1c* expression levels in mice with both MADM-induced UPD and in combination with genetic deletions (maternal, paternal, and cKO). These results are presented in Figure 2. Altogether, the data demonstrate that relative *Cdkn1c* expression in our experimental paradigms follows the pattern of expectation based on the imprinting of *Cdkn1c* and/or genetic conditional deletion.

2. Similarly, considering the unexpected results, the authors should show that is is *Cdkn1c* and not neighboring genes that are specifically impacted by the floxed deletion. This could be done globally via RNA-seq (preferred considering possible long-distance regulatory interactions) or locally via qPCR of nearby genes. Obviously, genes other than *Cdkn1c* could be secondarily impacted, which would be anticipated and in support of direct impact on *Cdkn1c* as long as *Cdkn1c* itself is altered as well.

We thank the reviewer for raising this important point. We have followed the reviewer's suggestion and performed RNA sequencing experiments in cortical cells in cKO and compared to control. We analyzed the genomic region located 2 Mbp upstream and downstream of *Cdkn1c* (including the entire *Kcnq1*-cluster of imprinted genes). Strikingly, the only gene that was significantly differentially expressed (i.e. downregulated) within the analyzed region was *Cdkn1c* itself. These data are presented in new Figure 3e.

Minor points:

1. Considering the differences between alleles, it would be useful to show the full gene/locus genomic structure for the loss-of-function allele in addition to the simple schematics shown in extended fig 1. This could be a supplemental figure.

We agree with the reviewer and now provide an overview with the two deletion alleles (Matsumoto et al. 2011; Zhang et al. 1997) (new Supplemental Figure 9). In addition we include a schematic summarizing the main findings (new Figure 6).

References:

- Furutachi S, Miya H, Watanabe T, Kawai H, Yamasaki N, Harada Y, Imayoshi I, Nelson M, Nakayama KI, Hirabayashi Y, Gotoh Y (2015) Slowly dividing neural progenitors are an embryonic origin of adult neural stem cells. *Nature neuroscience* 18 (5):657-665. doi:10.1038/nn.3989
- Mademtoglou D, Asakura Y, Borok MJ, Alonso-Martin S, Mourikis P, Kodaka Y, Mohan A, Asakura A, Relaix F (2018) Cellular localization of the cell cycle inhibitor *Cdkn1c* controls growth arrest of adult skeletal muscle stem cells. *eLife* 7. doi:10.7554/eLife.33337
- Mairet-Coello G, Tury A, Van Buskirk E, Robinson K, Genestine M, DiCicco-Bloom E (2012) p57(KIP2) regulates radial glia and intermediate precursor cell cycle dynamics and lower layer neurogenesis in developing cerebral cortex. *Development* 139 (3):475-487. doi:10.1242/dev.067314
- Matsumoto A, Takeishi S, Kanie T, Susaki E, Onoyama I, Tateishi Y, Nakayama K, Nakayama KI (2011) p57 is required for quiescence and maintenance of adult hematopoietic stem cells. *Cell stem cell* 9 (3):262-271. doi:10.1016/j.stem.2011.06.014
- Zhang P, Liegeois NJ, Wong C, Finegold M, Hou H, Thompson JC, Silverman A, Harper JW, DePinho RA, Elledge SJ (1997) Altered cell differentiation and proliferation in mice lacking p57KIP2 indicates a role in Beckwith-Wiedemann syndrome. *Nature* 387 (6629):151-158. doi:10.1038/387151a0

Reviewers' Comments:

Reviewer #1:

Remarks to the Author:

First of all I would like to congratulate Susanne Laukoter and colleagues for the number of convincing experiments added to the manuscript entitled "Unexpected role of imprinted *cdkn1c* genomic locus in cerebral cortex development". The revised version of text faced most of the observations previously made to the investigation. Although, I don't think that a complete correlation exists between mRNA levels and protein amounts, the evaluation of p57 transcript seems to me quite sufficient to confirm the various genetic handling. The probable identification of a growth-promoting function of *Cdkn1c*, although demonstrated in mice (and not in humans), and only in a specific tissue, is extremely promising. Moreover, it is in accord with the complexity of gene (and its regulation) and of the encoded protein itself. Probably, the author should have mentioned that the gene encoding mouse p57 is quite different from the human counterpart, and thus that the identification of a region in the mouse gene does not mean that it has the same function in humans. However, the possibility that a specific region of *Cdkn1C* gene correlates with a positive growth activation is novel and deserves additional studies in the future. But there is time for this.

Reviewer #2:

Remarks to the Author:

The general responsiveness of the authors to the criticism raised by the reviewers is impressive, with substantial new data supporting the manuscript. This is a thorough investigation of *Cdkn1c* imprinting and the effect of loss of parental alleles on cortical neurogenesis that, overall, is done in a rigorous manner. Despite the mainly productive efforts to improve the revised manuscript and the nicely laid out evidence for phenotypes in cKO cells and cortex, the key issues remain unanswered regarding the source of one of the primary findings. Specifically, there is an apparently paradoxical result that the MADM uniparental disomy situation still somehow maintains cell autonomous function while the conditional deletion allele does not. The presumed genomic "survival element" appears to be distinct from gene expression of *Cdkn1c*. I would not put such weight on requesting more robust demonstration of this, but the primary conclusions of the paper rest on whether this model is correct. As such, I feel that the authors must perform all basic experiments to rule out the possibility that some form of functional *Cdkn1c* transcript is produced in the UPD but not in the genetic ablation and cKO model.

My understanding is that this survival element can be mapped to a distinct genomic region that is different between the alleles used for different mouse lines and no other known genes in the locus show transcriptional differences in the cKO RNA-seq besides *Cdkn1c*. If this is correct, did the authors also look to see if there were unannotated transcripts within the disparate genomic survival element? Did the authors look for distinct unannotated transcripts within the *Cdkn1c* locus? The authors must definitively show that there is no evidence for any RNA difference between the UPD and ablation except for the documented changes in *Cdkn1c*. As the *Cdkn1c* RNA estimates for the MADM disruption are based on RNA-seq, the authors should generate a coverage plot for the RNA reads mapped to either strand across the locus. This should make it clear if there is something unexpected regarding unannotated or partial transcripts. If the authors can definitively show no RNA-based differences can explain this, there will still be a puzzle, but at least it won't be due to failure in the presumed imprinting of *Cdkn1c*.

Laukoter et al., Response to Reviewers - NCOMMS-19-09406A

We would like to sincerely thank the reviewers for their encouraging excitement and positive feedback to our revised manuscript. We have now further revised the manuscript based on the remaining open points mentioned in the reviewers' feedback. To this end we have furthered and deepened our analysis (new Supplemental Figure 3 and Reviewer Figure 1), and revised the discussion accordingly. All revisions/additions in the manuscript are highlighted in yellow. Below we address the reviewers' comments in a point-to-point response.

Point-to-Point Response (original reviewer comments are copied in blue):

Reviewer #1 (Remarks to the Author): First of all I would like to congratulate Susanne Laukoter and colleagues for the number of convincing experiments added to the manuscript entitled "Unexpected role of imprinted Cdkn1c genomic locus in cerebral cortex development". The revised version of text faced most of the observations previously made to the investigation. Although, I don't think that a complete correlation exists between mRNA levels and protein amounts, the evaluation of p57 transcript seems to me quite sufficient to confirm the various genetic handling. The probable identification of a growth-promoting function of Cdkn1c, although demonstrated in mice (and not in humans), and only in a specific tissue, is extremely promising. Moreover, it is in accord with the complexity of gene (and its regulation) and of the encoded protein itself. Probably, the author should have mentioned that the gene encoding mouse p57 is quite different from the human counterpart, and thus that the identification of a region in the mouse gene does not mean that it has the same function in humans. However, the possibility that a specific region of Cdkn1C gene correlates with a positive growth activation is novel and deserves additional studies in the future. But there is time for this.

We would like to thank the reviewer for the enthusiastic comments. We definitely agree with the reviewer that future studies will be very important to investigate the relevance and generality of our findings in different species, especially human. To take this notion better into account in our revised manuscript we have now added a respective statement in the *Discussion* section.

Reviewer #2 (Remarks to the Author): The general responsiveness of the authors to the criticism raised by the reviewers is impressive, with substantial new data supporting the manuscript. This is a thorough investigation of Cdkn1c imprinting and the effect of loss of parental alleles on cortical neurogenesis that, overall, is done in a rigorous manner. Despite the mainly productive efforts to improve the revised manuscript and the nicely laid out evidence for phenotypes in cKO cells and cortex, the key issues remain unanswered regarding the source of one of the primary findings. Specifically, there is an apparently paradoxical result that the MADM uniparental disomy situation still somehow maintains cell autonomous function while the conditional deletion allele does not. The presumed genomic "survival element" appears to be distinct from gene expression of Cdkn1c. I would not put such weight on requesting more robust demonstration of this, but the primary conclusions of the paper rest on whether this model is correct. As such, I feel that the authors must perform all basic experiments to rule out the

possibility that some form of functional *Cdkn1c* transcript is produced in the UPD but not in the genetic ablation and cKO model.

My understanding is that this survival element can be mapped to a distinct genomic region that is different between the alleles used for different mouse lines and no other known genes in the locus show transcriptional differences in the cKO RNA-seq besides *Cdkn1c*. If this is correct, did the authors also look to see if there were unannotated transcripts within the disparate genomic survival element? Did the authors look for distinct unannotated transcripts within the *Cdkn1c* locus? The authors must definitively show that there is no evidence for any RNA difference between the UPD and ablation except for the documented changes in *Cdkn1c*. As the *Cdkn1c* RNA estimates for the MADM disruption are based on RNA-seq, the authors should generate a coverage plot for the RNA reads mapped to either strand across the locus. This should make it clear if there is something unexpected regarding unannotated or partial transcripts. If the authors can definitively show no RNA-based differences can explain this, there will still be a puzzle, but at least it won't be due to failure in the presumed imprinting of *Cdkn1c*.

We thank the reviewer for encouraging and positive feedback. We very much agree with the reviewer that it is important (and critical with regard to our conclusions in the manuscript) to analyze if there were unannotated transcripts within the *Cdkn1c* locus. We also agree that we must definitely show that there is no evidence for any RNA difference between the UPD and ablation except for the documented changes in *Cdkn1c*. To address these critical issues we have followed the reviewer's suggestion and generated coverage plots for the RNA reads in the *Cdkn1c* locus as well as for the whole *Kcnq1* cluster. We thank the reviewer for this suggestion since RNA-seq is an unbiased method to detect transcripts throughout the genome. Thus any difference in the transcriptional output will be visible in such analysis. These new data are presented in the new Supplementary Figure 3 and below as Reviewer Figure 1 (which also includes all annotated as well as newly assembled transcripts, based on our RNA-Seq data, present in the *Cdkn1c* locus). Altogether, we now show in our coverage plots that there is no evidence for any RNA difference in all genetic paradigms, except for the expected differences in *Cdkn1c*. We also show that no un-annotated transcripts appear in our coverage plots in terms of transcriptional output from the *Cdkn1c* locus. Since the sequencing protocol used here does not preserve transcriptional orientation we have also performed reference annotation based transcript (RABT) assembly which is able to identify transcriptional orientation based on splice site orientation. This analysis has, as well, not revealed any novel transcripts. In conclusion, based on our comprehensive analysis (as documented in the new Supplementary Figure 3 and below in Reviewer Figure 1), we come to the following conclusions: 1) We can rule out the possibility that some form of functional *Cdkn1c* transcript is produced in the UPD but not in the genetic ablation and cKO model; 2) the absence of any novel transcript in the *Cdkn1c* locus provides additional evidence to corroborate our findings in the manuscript.

Reviewer Fig. 1

Reviewer Fig 1. RNA-seq read coverage plot and transcript annotations in *Cdkn1c* genomic locus. UCSC genome browser (<https://genome.ucsc.edu/>) display of RNA-seq read coverage as average size normalized coverage per base pair. **(a-f)** *Cdkn1c* locus position Chr7:143455000-143465000 (mm10/GRCm38). **(a-c)** FACS purified E16.5 MADM-7 (*MADM-7^{GT/TG};Emx1^{Cre/+}*) matUPD cells (a, 7 biological replicates), patUPD cells (b, 7 biological replicates) and matching control cells (c, 8 biological replicates). **(d)** E16.5 Control (7 biological replicates) and **(e)** cKO-*Cdkn1c*-MADM-7 (*MADM-7^{GT,Cdkn1c/TG,Cdkn1c};Emx1^{Cre/+}*) cells (3 biological replicates). **(f)** UCSC genome browser annotations: Gencode VM23 (top, blue), CpG island (middle, green), RepeatMasker (bottom, black). Note that reference annotation based transcript assembly, shown below each coverage track, did not show any signs or evidence of novel transcripts in the *Cdkn1c* genomic locus.

Reviewers' Comments:

Reviewer #2:

Remarks to the Author:

The authors have done everything reasonable to address the incongruous PatUPD vs cKO biological endpoints. I am applaud their work on this project. I recommend a couple changes to the presentation to put some of the author responses up front in the results and discussion, as I would guess that careful readers will be similarly perplexed.

1. Put the "Reviewer figure" showing Cdkn1c expression across the conditions in to the main figs - maybe within Panel 3e?
2. Change Lines 105-107 text from "show virtually no Cdkn1c expression" to "show drastic reduction of expression" or some similar relative statement than absolute as the data doesn't support "virtually no expression" as far as I can see.
3. Continuing on from the above point, I think that the PatUPD does look more robustly expressed compared to the cKO in the reviewer figure, though hard to judge based on the figure scaling alone. If it indeed the case that when comparing only cKO and PatUPD, the PatUPD is higher, it is worth mentioning this and that a small amount of transcription may escape imprinting in Pat UPD and there is the possibility that such transcription is sufficient for cell autonomous function. This could be handled in the discussion.

Responses to Reviewer's Comments (original comments are copied in blue)

Reviewer #2 (Remarks to the Author):

The authors have done everything reasonable to address the incongruous PatUPD vs cKO biological endpoints. I am applaud their work on this project. I recommend a couple changes to the presentation to put some of the author responses up front in the results and discussion, as I would guess that careful readers will be similarly perplexed.

We thank the reviewer for the encouraging and positive feedback. We appreciate that the reviewer is supportive and acknowledges our efforts. We now followed the suggestions for changes in the presentation. Textual changes are marked in yellow in the revised version.

1. Put the "Reviewer figure" showing Cdkn1c expression across the conditions in to the main figs - maybe within Panel 3e?

We thank the reviewer for this comment and have tried to incorporate the respective 'Reviewer figure' into the main Figures. However, due to the size of the data and because of space limitations (i.e. figure has to fit on single page according to the journals guidelines) it was not straight forward to incorporate the data into the main figure. Nevertheless, we agree that this piece of information is critical and elaborate in the main text about the results, and now also incorporated all data shown in the 'Reviewer figure' into the Supplementary Figures. Specifically, we have exchanged the upper panel of Supplementary Figure 3 (a-f) for the "Reviewer figure". Due to size constraints, former Supplementary Figure 3g-l is now presented as a separate figure (new Supplementary Figure 4). All successive supplementary figures have been renumbered accordingly and these changes are also reflected in the text.

2. Change Lines 105-107 text from "show virtually no Cdkn1c expression" to "show drastic reduction of expression" or some similar relative statement than absolute as the data doesn't support "virtually no expression" as far as I can see.

We have rephrased the statement accordingly to "show drastic reduction of expression".

3. Continuing on from the above point, I think that the PatUPD does look more robustly expressed compared to the cKO in the reviewer figure, though hard to judge based on the figure scaling alone. If it indeed the case that when comparing only cKO and PatUPD, the PatUPD is higher, it is worth mentioning this and that a small amount of transcription may escape imprinting in Pat UPD and there is the possibility that such transcription is sufficient for cell autonomous function. This could be handled in the discussion.

We thank the reviewer for pointing out this important issue. In order to more openly state that there is minimal expression from the silent paternal allele we now included a statement in the discussion.